# Intrinsic cell rheology drives junction maturation

K. Sri-Ranjan[1,6], J. L. Sanchez-Alonso [1,6], P. Swiatlowska[1], S. Rothery[1], P. Novak [2], S. Gerlach [3], D. Koeninger [3], B. Hoffmann[3], R. Merkel [3], M. M. Stevens [4], S. X. Sun [5], J. Gorelik [1]✉ & Vania M. M. Braga [1]✉

A fundamental property of higher eukaryotes that underpins their evolutionary success is stable cell-cell cohesion. Yet, how intrinsic cell rheology and stiffness contributes to junction stabilization and maturation is poorly understood. We demonstrate that localized modulation of cell rheology governs the transition of a slack, undulated cell-cell contact (weak adhesion) to a mature, straight junction (optimal adhesion). Cell pairs confined on different geometries have heterogeneous elasticity maps and control their own intrinsic rheology co-ordinately. More compliant cell pairs grown on circles have slack contacts, while stiffer triangular cell pairs favour straight junctions with flanking contractile thin bundles. Counter-intuitively, straighter cell-cell contacts have reduced receptor density and less dynamic junctional actin, suggesting an unusual adaptive mechano-response to stabilize cell-cell adhesion. Our modelling informs that slack junctions arise from failure of circular cell pairs to increase their own intrinsic stiffness and resist the pressures from the neighbouring cell. The inability to form a straight junction can be reversed by increasing mechanical stress artificially on stiffer substrates. Our data inform on the minimal intrinsic rheology to generate a mature junction and provide a springboard towards understanding elements governing tissue-level mechanics.

Epithelial cells are essential for multicellularity. The ability to adhere tightly to their neighbours and form boundaries within tissues assisted the diversification and specialization of different cell types, function integration and the evolutionary success of multicellular organisms. In an epithelial sheet, cells undergo fluctuations in cortical stiffness, cell density, area, and volume that underpin transitions from a rigid or jammed status to a less dense or fluid status[1–4]. These heterogeneous regions in the monolayer reflect areas of active cell rearrangements[5,6]. A central, but less understood aspect of this multidimensional, unifying model is the role that cell–cell contact plasticity plays in the coordinated response of neighbouring cells within tissues[2]. Junction remodelling is prominent in jammed-fluid transitions such as cell movement inside epithelial sheets, cell division, tissue morphogenesis, epithelial-to-mesenchymal conversion and collective cell motility[7].

A confluent epithelial monolayer contains densely packed, stiffer cells, with junctions that are more tense, stable, and mature. Mature junctions are under mechanical stress and can sense and generate pressure among neighbouring cells and across the epithelial monolayer[8–10]. In less dense regions, cells are more motile and are predicted to have relaxed junctions, compatible with migration

[1]National Heart and Lung Institute, Faculty of Medicine, Imperial College London, London, UK. [2]School of Engineering and Materials Science, Queen Mary University, London, UK. [3]Institute of Biological Information Processing, IBI-2: Mechanobiology, Julich, Germany. [4]Department of Materials, Department of Bioengineering and Institute of Biomedical Engineering Imperial College London, London, UK. [5]Department of Mechanical Engineering and Institute of NanoBioTechnology, Johns Hopkins University, Baltimore Maryland, USA. [6]These authors contributed equally: K. Sri-Ranjan, J. L. Sanchez-Alonso. ✉e-mail: j.gorelik@imperial.ac.uk; v.braga@imperial.ac.uk

(intraepithelial or otherwise)[6,11]. Such prediction is aligned with increased cell compliance that may accompany pathological status and lower functional competence, as shown in asthmatic epithelium[12] and during tumour progression. Indeed, tumour cells have higher deformability when compared to primary cells, a phenotype that is dependent on the transformation signal and correlates with metastatic potential[2–4,13]. Thus, cell biophysical properties play a remarkable influence on cell behaviour and (dys)function.

Cohesion in fully functional epithelium has characteristic features such as impermeability (via tight junctions), mechanical resistance (adherens junctions and desmosomes) and a hierarchical organization of signalling and structural components along their contacting interface[14]. Despite the relevance and wealth of information on cell–cell contact regulation, junction maturation per se is an ill-defined and poorly characterized process. As junctions mature, a unique remodelling of the actin cytoskeleton and signalling circuitry at cell–cell contacts takes place to generate a straight, taught junction, flanked by parallel thin bundles[15,16]. Apart from the typical configuration in epithelia as a straight cell–cell contact, there are no specific molecules that localize specifically or exclusively at mature junctions.

Several fundamental questions have not yet been elucidated. Cortical stresses have been extensively explored, but the contribution of intracellular rheology (i.e., viscoelasticity, cytoskeleton, organelles and nuclei)[17] is less understood: (i) how intrinsic cellular rheology influences the process of cell-cell contact maturation, and (ii) how neighbouring cells coordinate their viscoelastic and biomechanical responses at a joint interface to support stable adhesion. Importantly, a comprehensive model that integrates junction plasticity with mechano-responsiveness, cortical stiffness and viscoelasticity properties is currently unavailable. It is likely that dynamic changes in intracellular rheological properties of cells and their junctions feedback onto mechano-adaptation of epithelial cytoskeletal structures to reinforce and stabilize contacts[18]. At various intracellular sites, actin cytoskeleton mechano-adaptation responses have been shown to depend on the type of stress applied and to vary from reinforcement (increased cortical stiffness, stress fibres) to fluidization (i.e., depolymerization)[19,20].

Yet, the precise mechanisms that drive stress adaptive cytoskeletal reorganization that accompanies junction maturation have not been identified. It is unclear how localized biomechanical stress and junction maturation are generated at a common boundary shared by two neighbouring cells to ultimately determine monolayer organization. Part of the challenge has been how to assess specifically the mechanical profiles at mature junctions of epithelial monolayers. Informative data has been derived indirectly by measuring forces of cell attachment to E-cadherin as a substrate[21] or in suspended cells[22], molecular biosensors[23–27], traction forces on the substrate by epithelial colonies[2,28,29], the geometry of contacting membranes (vertex-based method)[30] or other techniques[8]. Thus, despite much progress on stress driven by or sensed at junctions, the specific contribution of intrinsic biomechanical stresses during cell–cell contact maturation has been underappreciated.

To shed insights into the coordination of cell behaviour in a monolayer, we investigate global cell mechanics and viscoelastic regulation of junction maturation and model the mechanical parameters that influence cell–cell contact morphology as a readout of junction plasticity. We designed a minimalistic system, the Junction Unit model, which contains a single junction between a cell doublet constrained on micropatterns of different geometries. Cell density, substrate stiffness and attachment area are kept constant. The uniqueness of our model is that surface stiffness and intracellular rheology are generated intrinsically by each cell in coordination with its neighbour, with no external forces applied by stretching or directly pulling/pushing at adhesion receptors. The model builds from seminal work using cell confinement that reveals how epithelial cell–cell contacts modulate the expression of differentiation markers[29,31], proliferation[32], directional migration[2,33], organelle distribution[34–37], and the influence of extracellular matrix attachment on junction positioning[38,39]. By addressing the intrinsic regulation of cell rheology, the Junction Unit model offers a unique opportunity to dissect the signalling and biophysical events leading to cell–cell contact maturation.

We find that a cell modulates its own rheology via fine-tuning the mechanical and viscoelastic properties in response to its neighbour and asymmetry (stress points or vertices of geometric shapes). Using an innovative adaptation of scanning ion conductance microscopy (SICM)[40,41], distinct high-resolution elasticity profiles spanning the whole doublet map precise stiffness hot-spots on the cell cortex with increasing stiffness depending on the micropattern geometry. In addition, we see marked differences in the recovery speed after elasticity probing. Such diverse intrinsic rheology generates remarkable types of junction morphology that resemble an immature, slack contact (as found in circular shapes) or a stable, mature junction (in triangular cell pairs). Our modelling and experimental data reveal the essential contribution of intracellular viscoelasticity, cortex stiffness and junction thickness underpinning the transition to straight, mature junctions. We unravel unexpected ways in which discrete localized changes in cell rheology modulate receptor density and actin dynamics during junction maturation as an adaptive, energy-dependent mechano-response.

Understanding how slack, unstable cell–cell contacts convert into mature junctions will highlight strategies to reinforce cell–cell adhesion and support appropriate epithelial differentiation and function in regenerative and pathological processes. The self-regulation of cell rheology in response to neighbouring cell pressure poses fascinating questions on how the mutual stimuli are coordinated and resolved into an equilibrium status: mature junctions with optimal configuration and maximum functionality. The framework described here will enable the introduction of further levels of complexity among multiple neighbours in a monolayer. Modelling of the integration of neighbour-derived viscoelastic stress propagation will inform mechanisms of tissue architecture, dynamics, and function.

## Results
### Geometric cell confinement alters cortex stiffness
A fundamental question in biology is how tight cohesion between neighbouring cells is maintained to enable signalling and resilience to chemical or mechanical stresses in different epithelial tissues. To investigate the involvement of cell rheology on the ability to attach to each other, we optimized a Junction Unit model, where a cell doublet attaches to different geometric shapes of the same area. Following optimization, we selected the best conditions for subsequent experiments as fibronectin-coated micropatterns with an area of 1300 μm² that contained well-spread cell doublets (Fig. 1a; Supplementary Fig. 1a, b). Neighbouring cells on circles had predominantly lamella at the periphery, while a high proportion of square or triangular cell pairs had thick F-actin fibres at their external borders (side fibres) and occasional short lamella (Fig. 1b, Supplementary Fig. 1c, d). The higher proportion of cells on squares and triangles containing stress fibre bundles suggested increased levels of intracellular mechanical stress compared to circular shapes. Such inference was supported by the longer distance between nuclei of cell doublets spread on triangles or squares (Fig. 1c).

We next evaluated how geometric confinement interferes with the morphology and viscoelastic properties of cell pairs[42]. In contrast to atomic force microscopy or other techniques, SICM does not touch the cell surface and thus interferes minimally with cell behaviour (Supplementary Fig. 2a–c). We used an updated version of SICM software that is appropriate to sense curvature to measure the height (see below). Cell height (distance bottom to cell apex) of cells on triangular shapes was significantly lower (Fig. 1d). The

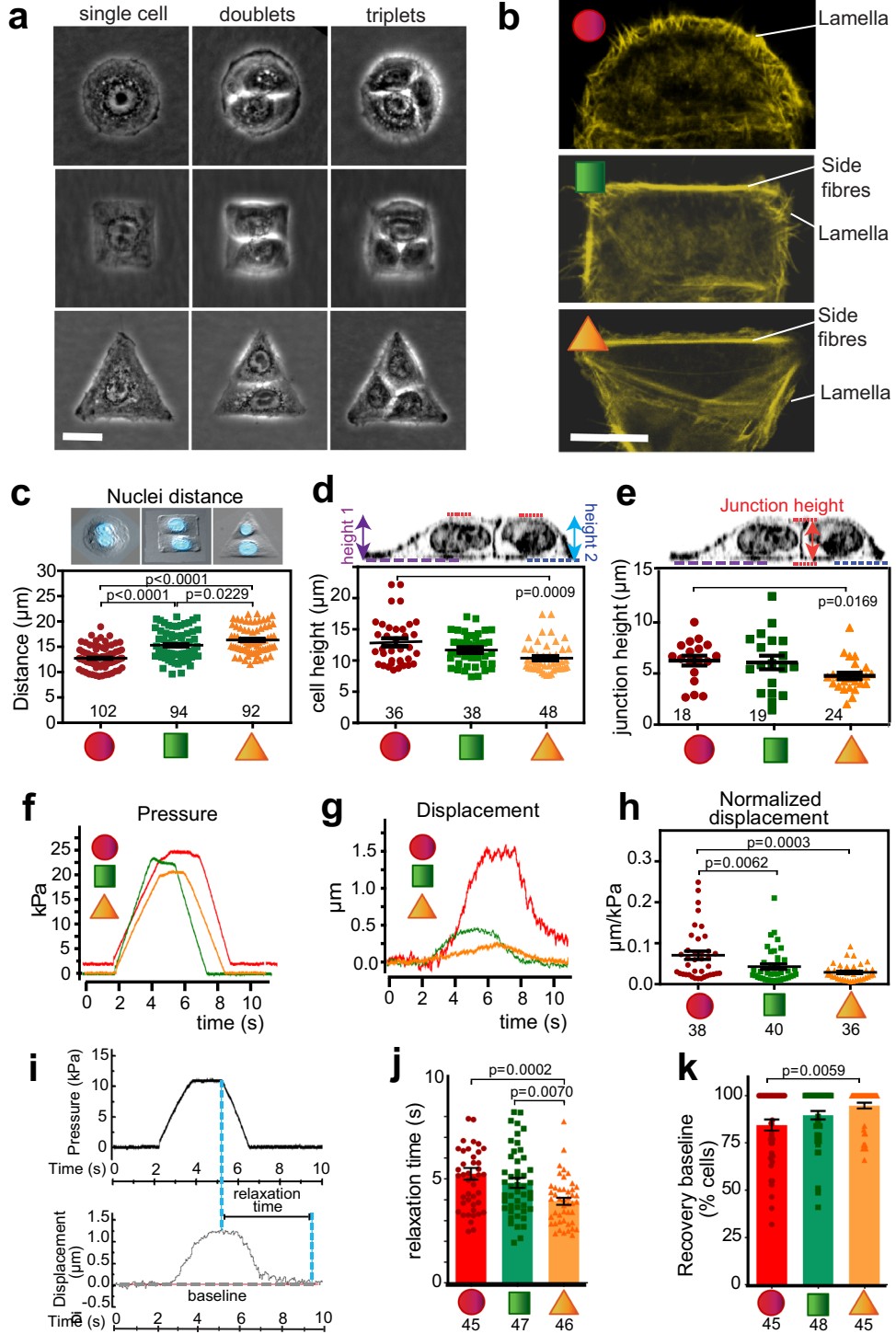

overall ability to form junctions was not impaired by higher mechanical stress, but junctions of triangular cell pairs were shorter (Fig. 1e). The time and spatial control of SICM pipette with constant fluid pressure on cell surfaces can also inform membrane displacement (Fig. 1f–h). Following displacement, faster membrane relaxation time and more efficient recovery to baseline were recorded on cell pairs in triangular or squared shapes (Fig. 1i–k). Those doublets also had stiffer cell apices, with a significant decrease in membrane displacement upon pressure (Fig. 1g, h). Our results strengthen the prediction that squared and triangular cell pairs show higher intracellular and cortical stiffness when compared to circular shapes.

## Global cell elasticity maps and biophysical parameters

Cell topography and a detailed elasticity map of the whole cell cortex of live cells were obtained simultaneously using an updated software suitable for distinct membrane curvature topography (Supplementary Fig. 3a, b; see the "Methods" section)[41,43]. Our work was based on Rheinlaender and Schäffer[44], which assumes a viscoelastic medium and measures the "pre-membrane cortex" stiffness, to a maximum vertical distance of 100 nm from the surface. While this methodology has been tested in other cell types, the precise correlation of Young's modulus values with known stiffer cellular structures has not yet been done.

High-resolution elasticity profiles clearly showed discrete variability of Young's modulus values along the surface of cell pairs (Fig. 2a).

**Fig. 1 | Geometric constraints impose different cortical stiffness in cell pairs.**
**a** Normal keratinocytes attached at random to fibronectin-coated micropatterns of different geometries. Representative phase contrast images were taken after 24 h growth. **b** Representative images of cell doublets stained with phalloidin to label F-actin showing distinct cytoskeletal structures: lamella on circular micropattern (red shape) and side fibres at cell peripheries of triangular (orange shape) or squared (green shape) micropattern. **c** Cell pairs were stained to visualize each nucleus and fluorescent and phase contrast images were acquired. Distance between nuclei on each micropattern was measured and sample numbers were shown inside the graph. **d–h** Cell pairs confined on different geometric shapes were scanned using scanning ion conductance microscopy (SICM) to measure cortical membrane topography upon pressure pulses (**d** and **e**) and membrane displacement (**f–h**). Measurements are shown for each geometric shape: cell height (bottom to the top of the nucleus) (**d**) and junction height (bottom to the top of contacting interface) (**e**). Numbers inside graphs show individual cell apex (**d**) or junctions (**e**). Representative trace of pressure applied at the cell apex (**f**), corresponding membrane displacement measured (**g**), and quantification of membrane displacement obtained on cell doublets on different geometries (**h**). **i** Graphs show membrane relaxation calculation as the time in seconds required to return to baseline upon pulse pressure release. **j** Average time required to relax the cortical membrane after applied pressure. **k** Percentage of cells on different micropattern shapes that recovered to a stable baseline after membrane deformation. Different micropattern geometries are represented at the bottom or inside graphs as red (circles), green (squares) and orange (triangles) lines or shapes. Numbers below geometric shapes in **h, j, k** show individual apex measured. Graphs represent average values between neighbouring cells sharing a micropattern from at least three independent biological replicates (thereafter $N = 3$). Mean values are shown and error bars represent the standard error of the means (thereafter SEM). Statistical analysis was performed one-way ANOVA followed by Kruskal–Wallis post-hoc test with Benjamin, Krieger, and Yekutieli test to control for multiple comparisons (**c–e, h, j, k**). Scale bars are 20 μm. Source data are provided as a Source Data file.

Stiffer hot spots highlighted precise areas where higher stress is predicted: cell apex and cell periphery (vertices or lamellae, Fig. 2a, b). Strikingly, on each geometric shape, mechanical stiffness at the junctional region was lower than in other areas measured (Fig. 2b). Furthermore, circular cell pairs had lower cortical stiffness at the cell apex and vertices/lamellae, while triangular-shaped cell doublets had the highest levels (Fig. 2b). There were moderate correlations between Young's moduli measured at junctions and cell apex or vertices in individual cells, irrespective of the geometry (Fig. 2d–e).

There was a strong positive correlation between the area and volume of each cell, but this effect was geometry-independent (Fig. 2f). In contrast, cell height did not seem to be strongly influenced by the area or volume measured in the same cell (Fig. 2g, h). Unexpectedly, Young's modulus values measured at the apex or vertices were not correlated with the respective height, volume or area of each cell (Supplementary Fig. 4). Because one cell can only grow at the expense of its neighbour on the confining micropattern, it is possible that shape, height, and volumes are regulated in a coordinated, dynamic fashion between neighbours. If this prediction is correct, then a clear dissimilarity in biophysical parameters should be detected between neighbouring cells sharing a micropattern.

## Cells sharing a confined area are not equal

Our preliminary evaluation showed that neighbouring cells shared the 1300 μm² micropattern with an average difference of 150–200 μm² between them. The largest significant difference was observed between circular and triangular cell pairs, while the ratio between the areas of cells on a micropattern was similar in all geometries (Supplementary Fig. 4g, h). A pairwise analysis was performed, where biophysical measurement in each cell was matched with its neighbour on different geometric micropatterns (Fig. 3a–d). The averages of smallest and largest cell area, volume or Young's modules between neighbours were significantly distinct in all geometric shapes (Fig. 3a, b, d). In contrast, circular cell doublets did not have significant differences in cell height (Fig. 3c). Cell height ($p < 0.025$) differences were affected by cell pair shape, while stiffness, area and volume differences did not vary significantly among geometries. Using Wilcoxon matched pair test, a comparison of matched cells sharing a micropattern showed that all differences were statistically significant (not shown). The unevenness between neighbouring cells suggests an inherent and dynamic behaviour of each cell sharing a confined space.

One of the consequences of the unequal cell size was that the boundary between the two cells was not always found in the middle of each micropattern. Rather, there was a preference for where junctions were placed (Fig. 3e). On circular shapes, junctions were mostly located at the equator, while a high percentage of junctions were spanning the sides of squared and triangular cell doublets, providing equal or unequal area partition between neighbours. However, the random chance that a junction was found at a given 5° degree interval was 2.78% in circular shapes. In contrast, there was a higher likelihood of a junction spanning a vertex of squares or triangles than their sides. The potential reasons for the orientation preference are currently being investigated in the lab.

Furthermore, the configuration of junctions was distinct, from a wave morphology in circular cell pairs to a straighter junction of neighbours on a triangular shape (linearity index closer to 1; Fig. 3f). Junctions found in triangular cell pairs were also significantly shorter in length (Fig. 3g). Yet, E-cadherin staining occupied similar length of the available contacting interface between cell pairs on different geometries (Fig. 3h). We surmised that the unequal biophysical properties of cells sharing a micropattern (Fig. 3a–d) may affect qualitatively and quantitatively the way they interact with each other (Fig. 3e–h).

## Modelling of the impact of biophysical parameters on junction configuration

The biological data showed that cell pairs on circular shapes with the highest compliance have the most curvilinear-shaped junctions, while stiffer cell pairs (i.e., on triangular shapes) have straighter and shorter junctions (Fig. 3f). Previous modelling available in the literature focuses on cell motility, spreading or the behaviour of cells confined or within colonies/sheets in a dynamic or static condition (e.g. refs. [18,45–47], and references therein). While their contribution to our understanding has been considerable, these models did not consider the specific questions we address here, i.e., the interplay between cell rheology and junction properties. To obtain insights into the impact of the cell biophysical properties on cell–cell adhesion, we modelled the cell cortex as a viscous fluid layer that can generate active contractile force (Fig. 4)[48,49]. Using static images of a single junction from cell pairs with different rheological properties, the model uses a Young–Laplace equation but considers a mechanical equilibrium for shape calculation. It also considers that tension is balanced by active stress modulated by cells together with internal pressure (see ref. [49] for discussion and Supplementary Note). The cell–cell junctional interface is connected by E-cadherin molecules modelled as elastic springs. For each cell surface, the mechanical stress in the actomyosin network at the cortex is balanced with the pressure ($P$) in the cytoplasm (e.g., hydraulic or viscoelastic pressure) as well as elastic forces in the E-cadherin bonds ($k$) on the extracellular side of the membrane (Fig. 4a). The parameter ($k$) is related to the stiffness of E-cadherin bonds and their density: it reflects a particular spatial distribution and number of E-cadherin bonds based on the E-cadherin intensity from experimental data. We assume that the E-cadherin bonds are generally perpendicular to the cell surface,

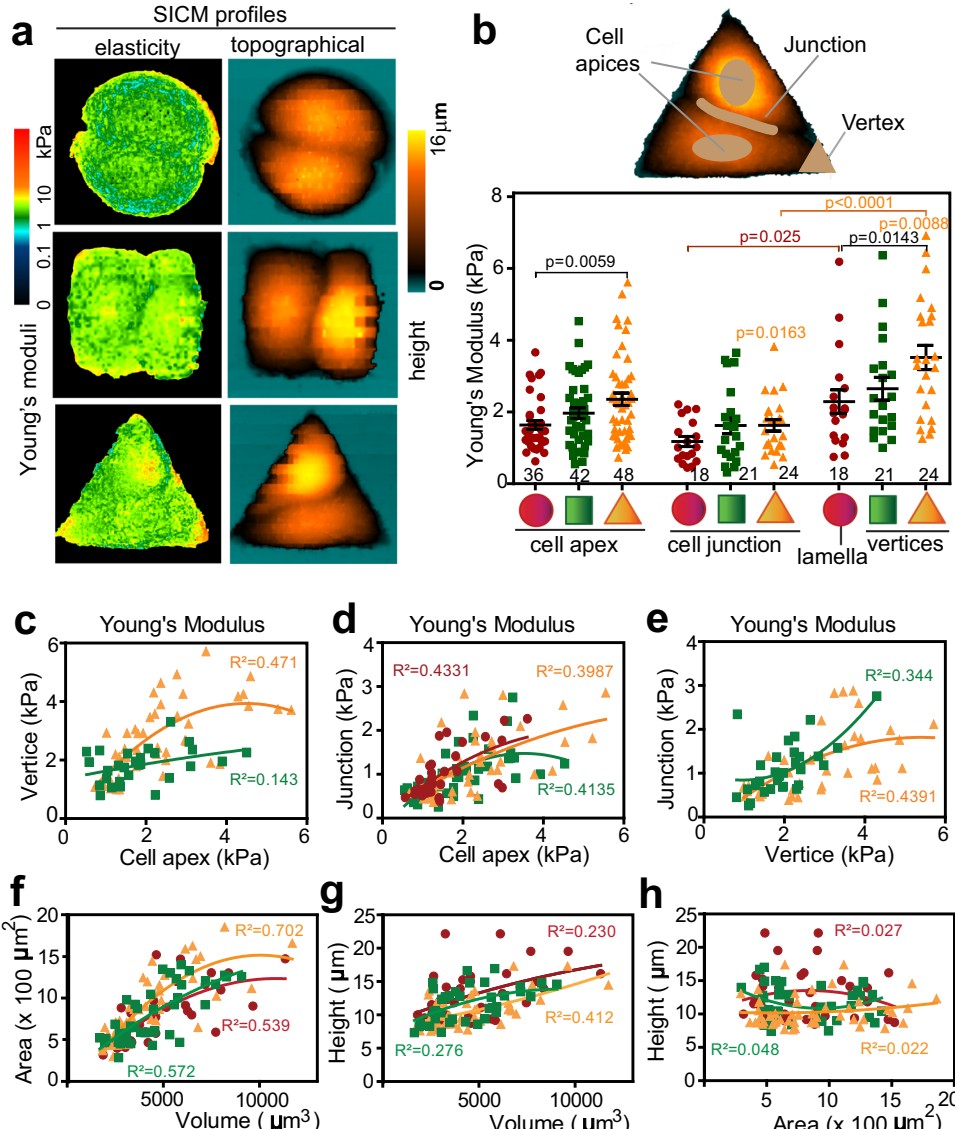

**Fig. 2 | Elasticity profiles and correlation of biophysical parameters under various geometric constraints. a** Cortical elasticity map. Representative SICM images of cell pairs showing elasticity (Young's moduli, left) and topography (right) maps that were obtained simultaneously. Colour scale bar on the left represents Young's modulus values of the stiffness maps from softer (dark blue) to stiffer (red). On the right, the colour scale bar shows the height from baseline (black) to cell top (yellow). **b** Diagram shows selected cellular regions of confined cell pairs that had their stiffness measured. Graph shows Young's modulus values plotted per geometric shape and grouped per cell region. **c**–**e** Correlation of stiffness of individual cells measured at the cell apex, at vertices (**c**) or junction of a cell pair (**d**). Stiffness measured at vertices or junctions was also correlated (**e**). **f**–**h** Individual cell volume, height and area of keratinocytes grown on micropatterns were measured

with SICM and plotted to determine their correlation: individual cells were measured on circles ($N = 36$; squares $N = 39$–$42$; triangles $N = 48$). Goodness of fit ($R^2$) values are shown for each comparison and geometric shape. Different micropattern geometries are represented at the bottom or inside graphs as red (circles), green (squares) and orange (triangles) lines or shapes. Scan sizes shown in (**a**) are: circles 40 µm × 40 µm, squares 46 µm × 46 µm and triangles 53 µm × 53 µm. Graphs represent average values between neighbouring cells sharing a micropattern and error bars are SEM ($N = 3$ (**a**, **b**); $N = 7$ (**c**–**h**). Statistical analysis was performed by one-way ANOVA followed by Kruskal–Wallis post-hoc test and Benjamin, Krieger and Yekutieli test. Source data are provided as a Source Data file. $p$ values in black font are comparisons across shapes; $p$ values in coloured font represent comparisons within a geometric shape.

and thus, the force balance in the normal direction for each cell surface at the interface reads:

$$\begin{cases} P_1 - f_1 H_1 + k\epsilon = 0 \\ P_2 + f_2 H_2 + k\epsilon = 0 \end{cases} \tag{1}$$

where $P_i$ is the mechanical and viscoelastic pressure inside each cell, $H_i$ is the mean curvature of each cell boundary and $i$ labels cell 1 or cell 2 sharing a micropattern. The parameter $f_i$ is the combined cell surface tension. It is defined by ($f_i = h_i \sigma_i + T_i$); it includes the cell membrane tension $T_i$, cortical thickness ($h_i$) and the cortical stress $\sigma_i$ (Fig. 4a), which is tangential to the membrane. $\epsilon$ is the mechanical strain on the

E-cadherin bonds, described as $\epsilon = \frac{l - l_0}{l_0}$, where $l$ is the length of E-cadherin at junction and $l_0$ is the length of E-cadherin without force loading.

Given the mechanical force balance conditions of the cell boundaries, the shape of the cell–cell interface can be obtained by solving $H_i$ in Eq. (1). With the assumption that E-cadherin bonds are perpendicular to the central line between the two membranes as shown in Fig. 4a, the individual cell boundaries can be written as

$$\left( \vec{r}_1(s) = \vec{r}(s) + \frac{l}{2}\vec{n} \right) \text{ and } \vec{r}_2(s) = \vec{r}(s) - \frac{l}{2}\vec{n},$$

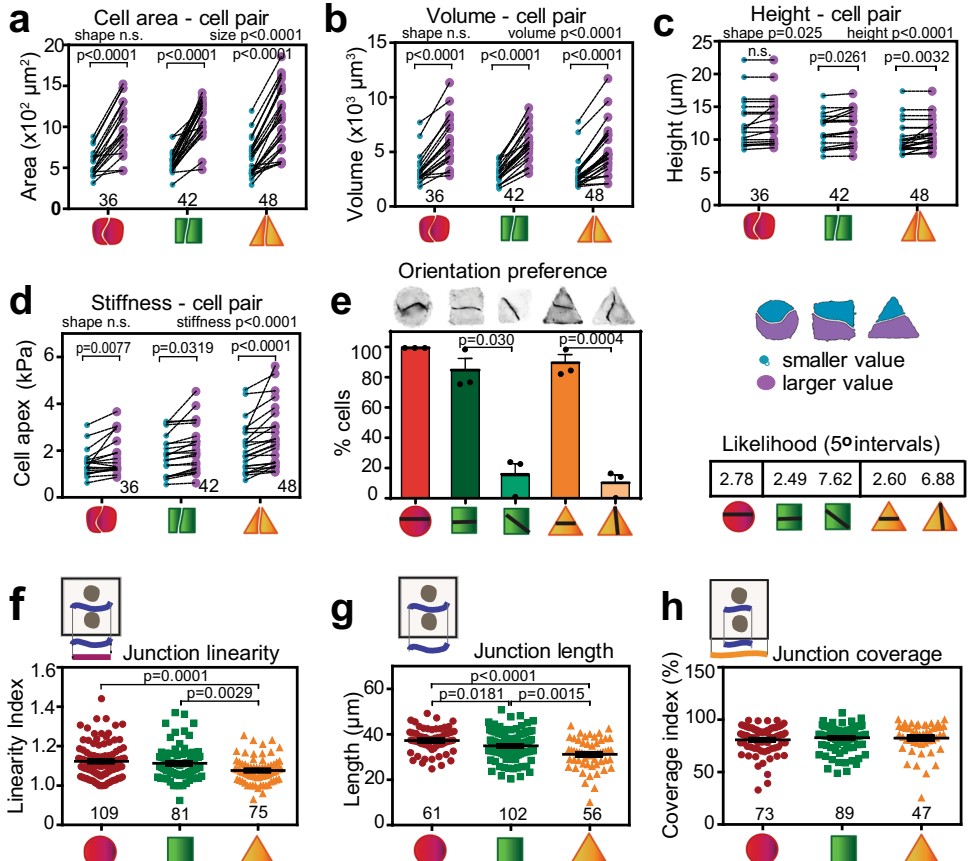

**Fig. 3 | Cells sharing a micropattern are unequal.** Keratinocytes grown on different geometries had their biomechanical parameters measured in each neighbour of a cell pair and ranked by value. Graphs show the pair-wise distribution with the value of neighbours sharing a micropattern linked by a black line: **a** cell area; **b** cell volume; **c** cell height; **d** Young's modulus measured at the cell apex. **e**–**h** Cells were stained for E-cadherin and F-actin and various parameters were quantified: the preference of junction positioning within micropatterns as a percentage of cells with a given orientation of cell-cell contacts (orientation preference), and their probability to be found at a 5° angle at the sides or vertices (likelihood, **e**); how straight junctions are (linearity index, **f**); the length of junctions (junction length, **g**); the percentage of the contacting interface area covered with cadherin receptors (junction coverage, **h**). Below graphs, micropattern geometries are represented as red circles, green squares and orange triangles. Number of independent biological replicates are: $N = 4$ (**a**–**d**); $N = 3$ (**e**); $N = 7$ (**f**–**h**). Number of cell pairs assessed is shown on top of each geometric shape (**a**–**d**; **f**–**h**). Mean values and error bars (SEM) are shown (**e**–**h**). Statistical analysis was performed by $t$-test with Welch's correction (unpaired, two-tailed, **e**) or one-way ANOVA followed by Kruskal–Wallis post-hoc test and Benjamin, Krieger and Yekutieli test (**f**–**h**). In panels **a**–**d**, two-way ANOVA test was performed followed by Sidák's multiple comparisons to test for interactions. Similarly, Wilcoxon matched paired test for each cell pair comparison showed significant differences for all samples (values shown in Source data). Source data are provided as a Source Data file.

where $\vec{r}(s)$ defines the central line describing the average cell–cell interface, $s$ is the length of the central line (curvilinear coordinate) and $\vec{n}$ is the unit vector perpendicular to the central line (Fig. 4a). Because the length of the E-cadherin bonds is very small when compared to the radius of curvature of the cell, we naturally have the condition $Hl \ll 1$. Through zero-order approximation, the curvature of the membranes of neighbouring two cells ($H_1$ and $H_2$) can be written as

$$H_1 = H + \frac{l''}{2} \text{ and } H_2 = H - \frac{l''}{2}$$

where the curvature of the central line is $H$. The governing equation for $l$ is

$$l'' - \left(\frac{1}{f_1} + \frac{1}{f_2}\right) k \frac{l - l_0}{l_0} - \left(\frac{P_1}{f_1} + \frac{P_2}{f_2}\right) = 0 \qquad (2)$$

where the boundary conditions are $l|_{s=0} = l|_{s=s_o} = l_0$, and $s_0$ is the central line measuring the total length of the boundary without mechanical load. Then Eq. (1) becomes linear equations for $l$ and $H$ and are solved numerically using the finite difference method. Based on the solution for $l$ in Eq. (2) the shape of the two cell boundaries ($H_1, H_2$) and

the central line ($H$) is given by

$$
\begin{aligned}
H_1 &= \frac{1}{f_1}\left(P_1 + k\frac{l - l_0}{l_0}\right) \\
H_2 &= -\frac{1}{f_2}\left(P_2 + k\frac{l - l_0}{l_0}\right) \\
H &= \frac{H_1 + H_2}{2}
\end{aligned}
\qquad (3)
$$

Various scenarios to alter the configuration of cell–cell interfaces were modelled as curved or sinusoidal shapes as observed experimentally (Fig. 4b–g). By increasing the intrinsic hydraulic pressure ($P$) in one cell, the model predicts that the membrane interface would bulge into the cell that has smaller pressure (Fig. 4c, f). The amplitude of the interface curvature could be reduced by re-calibrating the intrinsic pressure (Fig. 4c) or increasing the cortical stress $\sigma$; a key contributor to the parameter $f_i$ in Eq. (1) (combined cell surface tension); Fig. 4f). A sinusoidal interface shape can be generated and refined in the same way, except that a gradient, localized variation of the intrinsic pressure at the interface is predicted (Fig. 4b, e). The latter implies that there should be a potential flow of material along the contacting interface, consistent with the flow and heterogenous

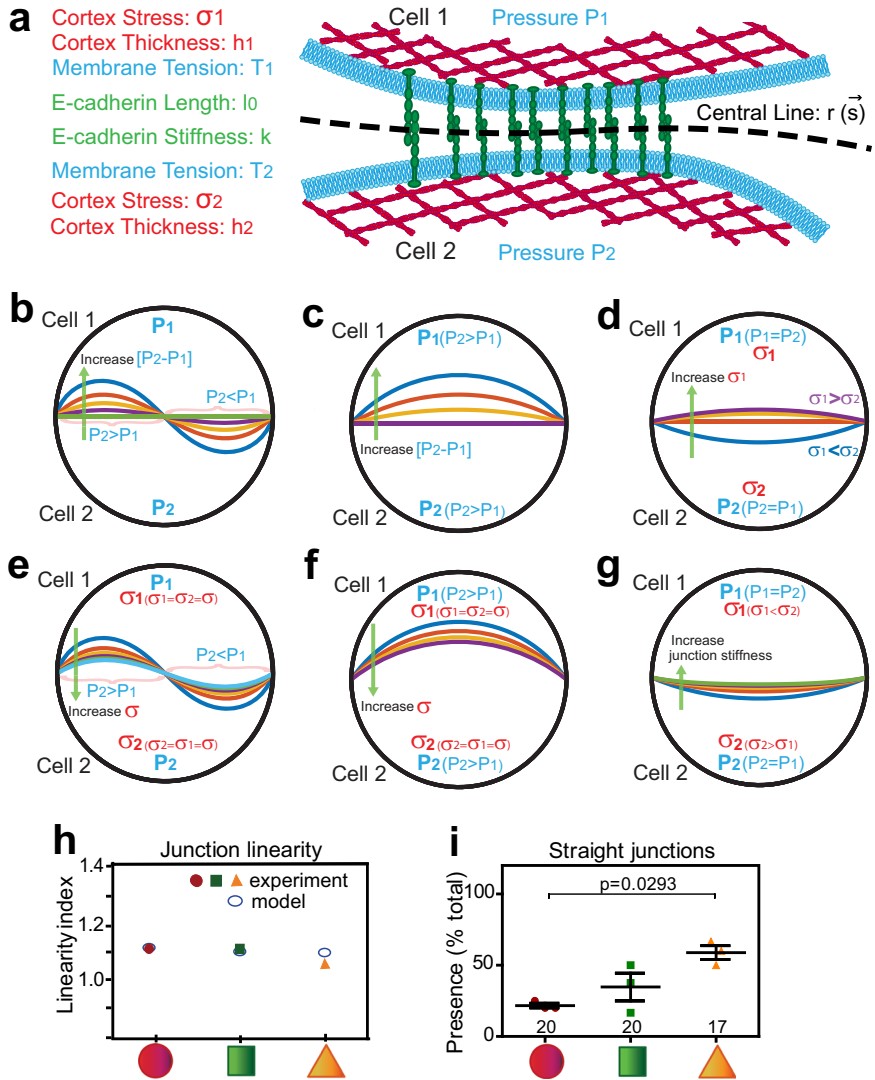

**Fig. 4 | Modelling the role of viscoelasticity properties in determining junction configuration and maturation. a** Diagram showing junction representation and parameters used in the modelling of junction configuration (curvilinear or straight) under different elasticity conditions. Zoom of the junctional region showing the parameters for cell 1 and cell 2 described in the model. **b–g** Distinct scenarios in which a curvilinear or sinusoidal junction appears and can be straightened up are shown by lines of different colours and progressive curvature. Distinct intracellular pressure between neighbours is predicted to bend their contacting interface, which can be reversed by various mechanisms. **b, e** Under conditions of constant intracellular pressure ($P_1$, $P_2$), discrete, localized variations of pressure from cell 1 or cell 2 leads to an undulated or "wavy" contacting interface. The latter can be straightened up by local realignment of intracellular pressure (**b**) or increasing cortex stress $\sigma$ (**e**). **c, f** Unequal global intracellular pressure among cells ($P_2 > P_1$) results in a bulging, curvilinear junction. The interface curvature can be counteracted by reducing the difference in intracellular pressure between neighbours (**c**) or an

increase in the overall cortex stiffness ($\sigma$) of the cell pair (**f**). **d, g** Under conditions of similar pressure but unequal cortex stiffness between neighbouring cells ($\sigma_1 < \sigma_2$), a straight junction is obtained by rebalancing the cortical stiffness in one cell (**d**) or increasing junction stiffness (**g**). Different coloured interface curvatures in the diagrams represent the remodelling of contacting interfaces upon changes in the respective biophysical parameter. **h** Model fitting of junction linearity. The graph shows the linearity index (length of cadherin junctions over contacting interface) of experimental data obtained herein and values predicted by our modelling.
**i** Quantification of the percentage of micropatterns in which cells display straight junctions by visual inspection. Below graphs (**h** and **i**), micropattern geometries are represented as red circles, green squares and orange triangles. Number of cell pairs quantified is shown on top of each geometric shape ($N = 3$). Mean values and error bars (SEM) are shown in (**i**). One-way ANOVA followed by Kruskal–Wallis post-hoc test with Benjamin, Krieger, and Yekutieli for multiple comparisons (**i**). Source data are provided as a Source Data file.

organization of receptors, F-actin, and membrane along cell–cell contacts shown previously[10]. Yet, the current model did not compute these dynamic changes in the tangential direction of the interface but was calculated from static images, at a mechanical equilibrium using force balance in the normal component.

Even when the intracellular pressure is equal between neighbouring cells, the model predicts that curvature of cell–cell contacts could be obtained by unequal cortical stress between two cells ($\sigma$, Fig. 4d, g). When differences in cortical stress between neighbouring cells exist, a shallower curvature could be achieved by two means: increasing the cortical stress of the most compliant cell

(Fig. 4d) or stiffening the bonds of E-cadherin receptors ($k$; Fig. 4g). Thus, for unequal cells sharing a confined space (Fig. 3), the distinct rheological pressures (cortex, intracellular viscoelasticity and junctions) impact on the shape and configuration of cell–cell contacts.

Furthermore, or alternatively, a contributor to curvilinear contacts could be the nuclei of the cell pair, which were found closer to cell junctions in the more compliant circular shapes (Fig. 1c). The consequence of the proximity of large nuclei could be to generate steric hindrance to the organization of junctional structures in the cytoplasm that strengthen adhesion. The mechanical variables can be derived from experimental data, assuming conditions that: (i) the cytoplasmic

viscoelastic pressure is the same in the two cells (i.e. Fig. 4d, g) and therefore balances each other and (ii) the pressure on the cell apex is mainly from the nucleus. As cortical tension is generally much higher than membrane tension, therefore membrane tension $T_i$ can be ignored (Fig. 4a).

The pressure in the cell may be a combination of hydraulic pressure in the cytoplasm and mechanical pressure from the cell nucleus impacting the cell-cell boundary. To estimate the mechanical variables from the images, we can write:

$$P_1 = \frac{\alpha_1}{ND}, P_2 = \frac{\alpha_2}{ND}, f_1 = \alpha_3 I_{actin}, f_2 = \alpha_4 I_{actin}, k = \alpha_5 I_{Ecad}, l_0 = \alpha_6$$

where ND is the distance of the nucleus from the position along junctional interface; $f_1, f_2$ are cortical stress of each cells; $I_{actin}, I_{Ecad}$ are intensity of F-actin and E-cadherin, $l_0$ is the length of E-cadherin without force loading, $\alpha_1$ to $\alpha_6$ are constant and independent of $s$, the arclength along the cell-cell junction line $r$. They are fitting parameters that relate the experimental image data to the mechanical variables (i.e., $\alpha_1$ refers to the force variable, while $\alpha_6$ is the junction length variable). Because of calibration, i.e., direct recasting into physical parameters is not possible, only relative changes in the parameters can be interpreted. These fitting parameters depend on the experimental condition and the type of cell in question.

We can compute the junction shape as above for a given set of $\alpha_1$ to $\alpha_6$ and find their best fitting values upon variation of selected parameters used in the model (Supplementary Fig. 5). From experimental data quantifications, we can extract the values for E-cadherin intensity, F-actin intensity, boundary length and nuclear distance as input to the model. Among the 6 coefficients, only 4 of them are independent ($\alpha_2$ to $\alpha_5$). Therefore, we fix $\alpha_1 = 1$ and $\alpha_6 = 0.001$ and only adjust $\alpha_2$ to $\alpha_5$ based on data of cell pairs on different geometric shapes and compare the output of the model versus the linearity measured experimentally. The graph obtained from the modelling incorporating nucleus contribution generated values very close to the experimental data on linearity index (Fig. 4h). The latter was confirmed with the qualitative inspection of the percentage of junctions with straight configuration: higher tensile cell pairs have straighter junctions (Fig. 4i). Therefore, the model shows (Fig. 4; Supplementary Fig. 5) that the straightness of the interface between two cells is determined by the combined contribution of intracellular mechanical and viscoelastic pressure, cortical stiffness (cortical stress and thickness), junction stiffness (E-cadherin strength and cytoskeletal structures at junctions), and nucleus positioning. These contributions were then tested experimentally.

## Stiffer cells contacts mimic mature epithelial junctions

We next validated whether cell pairs on the highest tensile geometric shape have a bone fide organization as observed in mature junctions seen in epithelial monolayers. All cell pairs had junctional actin that co-localized with E-cadherin-labelled contacts, irrespective of the micro-pattern geometry (JA, Fig. 5a–c). In contrast, the presence of parallel thin bundles at junctions typical of mature epithelial junctions was favoured in stiffer cells sharing squares or triangles (TB, Fig. 5b, d). In addition, a higher proportion of cells had thin bundles labelled with the contractile marker phosphorylated regulatory myosin light chain (PMLC; Fig. 5e, g). In contrast, independently of the cell pair shape and cortical stiffness, junctional actin staining with PMLC was discrete and in a reduced proportion of cells (Fig. 5e, f). Thus, in the Junction Unit model, F-actin networks at junctions respond to rheological pressures primarily by adaptation of thin bundles and their contractile properties, thereby resembling the typical F-actin structures of epithelial monolayers.

We demonstrated that the distinct F-actin organization of more tensile neighbours correlates with the higher Young's moduli

measured at different cell areas by SICM. We inferred that higher stiffness of triangular cell pairs (Fig. 2a, b) would drive increased levels of E-cadherin at junctions to enable cells to sustain the higher mechanical load at cell-cell contacts. However, contrary to our prediction, levels of cadherin receptors and F-actin at junctions were significantly reduced in squared or triangular cell pairs that have the highest resistance to deformation (Fig. 6a–e).

These results were unexpected, as there is no increase in receptor or F-actin levels to balance the mechanical stress per unit of cell-cell contact on those geometries. It is feasible that, rather than higher receptor density, intrinsic rheological pressures could trigger junction re-enforcement via distinct properties of the junctional actin pool and adjacent contractile thin bundles (Fig. 5e–g). In accordance with our modelling (Fig. 4), both junctional actin and thin bundles would contribute to the parameters cortex thickness ($h$) and stress ($\sigma$) at junctions (Fig. 3f). The corollary is that by changing the organization and properties of distinct F-actin pools, junction reinforcement and stiffness could be achieved without increasing the number of adhesive receptors.

To address the above premises, fluorescence recovery after photobleaching (FRAP; Fig. 6f–h) showed that junctional actin dynamics were altered in stiffer cells on squares or triangles. Following bleaching, maximum recovery of GFP-actin at junctions was similar in all geometric shapes (Fig. 6g). However, the time needed for fluorescence recovery was significantly longer in doublets grown on squares or triangles (Fig. 6h), indicating a more stable F-actin pool (Fig. 6c, h) driven by increased tensional constraints. The data support our model predictions that cells counterbalance their intrinsic mechanical and viscoelastic pressures at contacting interfaces via junctional actin stabilization (Fig. 6h) and thin bundles contractility (Fig. 5e–g), rather than increasing localized E-cadherin and F-actin levels. In turn, such mechanical adaptation supports a straighter conformation of junctions like those within confluent epithelial sheets.

## Intracellular mechanics as a driver of junction maturation

We next validated the role of intrinsic cell rheology during junction maturation. Following treatment with Y27632, an inhibitor of the Rho GTPase effector ROCK1, elasticity maps showed an overall reduction of Young's moduli and cell height of cells seeded in all geometries (Fig. 7a, Supplementary Fig. 6a). The cell apex stiffness was significantly lower only in squared or triangular cell pairs (Fig. 7b), while junctions became less stiff and flatter specifically in triangular doublets (Fig. 7c, d). Circular cell pairs were unresponsive to contractility inhibition (Fig. 7b–d), suggesting that Young's modulus of circular doublets is at a minimum. Consistent with this interpretation, following Y27632 treatment, nuclei collapsed towards junctions of squared and triangular pairs but had no effect on circular cell pairs (Supplementary Fig. 6b, c). We surmise that relaxation of intrinsic stiffness (i) abrogates the mechanical stress of squared and triangular pairs to levels similar to those on circular pairs and (ii) impairs the ability to localize the nucleus in the middle of a cell.

Upon treatment with Y27632, the correlation between cell apex stiffness and volume (Fig. 7e) or area (Supplementary Fig. 6d) of each cell remained poor for all geometric shapes. There was a clear reduction of cell volume and enlargement of cell area on squared or triangular shapes (Fig. 7f). Furthermore, inhibition of intracellular stiffness did not alter the strong correlation between area and volume of each cell (Fig. 7f). A positive correlation between cell area and volume was not observed in confined or unconstrained single cells published elsewhere[50]. This discrepancy could be due from the distinct cell types and methodology used. It may also reflect an equilibrium between two cells sharing a constrained surface, which would prevent cell spreading that accompanies relaxation.

We next addressed the question whether relaxation reverses the mature junction status in triangular cell doublets to a more immature

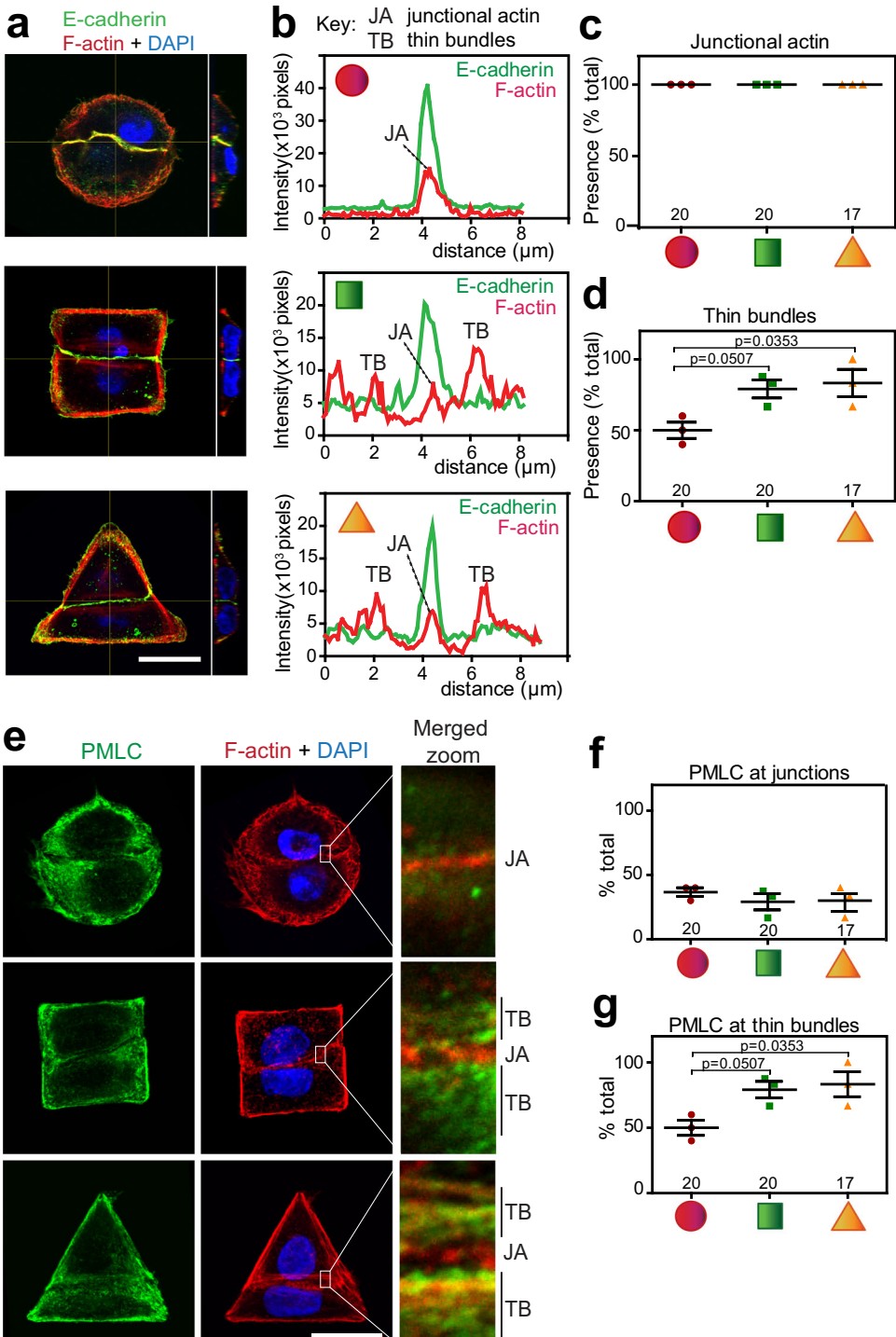

**Fig. 5 | Geometric constraints modulate the optimal organization of the epithelial cytoskeleton.** Keratinocytes grown on different geometries were fixed and stained for F-actin, nucleus, and E-cadherin (**a**) or phosphorylated myosin light chain (PMLC, **e**). **b** Representative line scan perpendicular to and at the middle of each junction was drawn and the intensity profile of E-cadherin (green) and F-actin (red) was obtained. Peaks of high intensity show the junctional actin (JA) co-localizing with E-cadherin. Parallel, adjacent thin bundles (TB) are clearly seen in squared and triangular cell pairs. **c** and **d** Percentage of micropatterns showing the presence of junctional actin (**c**) or thin bundles (**d**) across different geometries. **e** Representative images show the distribution of contractile filaments (PMLC,

green) of cells seeded on different geometric shapes. Merged zoom of junctional areas on the right highlights the presence of junctional actin (JA) and/or thin bundles (TB). **f** and **g** Graphs show the percentage of cell pairs with PMLC labelling at junctional actin (**f**) or thin bundles (**g**). Below graphs (**c**, **d**; **f**, **g**), micropattern geometries are represented as red circles, green squares and orange triangles. Confocal projections are shown (**a**, **e**). Scale bar = 20 µm. Number of micropatterns quantified is shown on top of each geometric shape. $N = 3$. Mean values and error bars are shown (SEM). Statistics used one-way ANOVA followed by Kruskal–Wallis post-hoc test and Benjamin, Krieger and Yekutieli test. Source data are provided as a Source Data file.

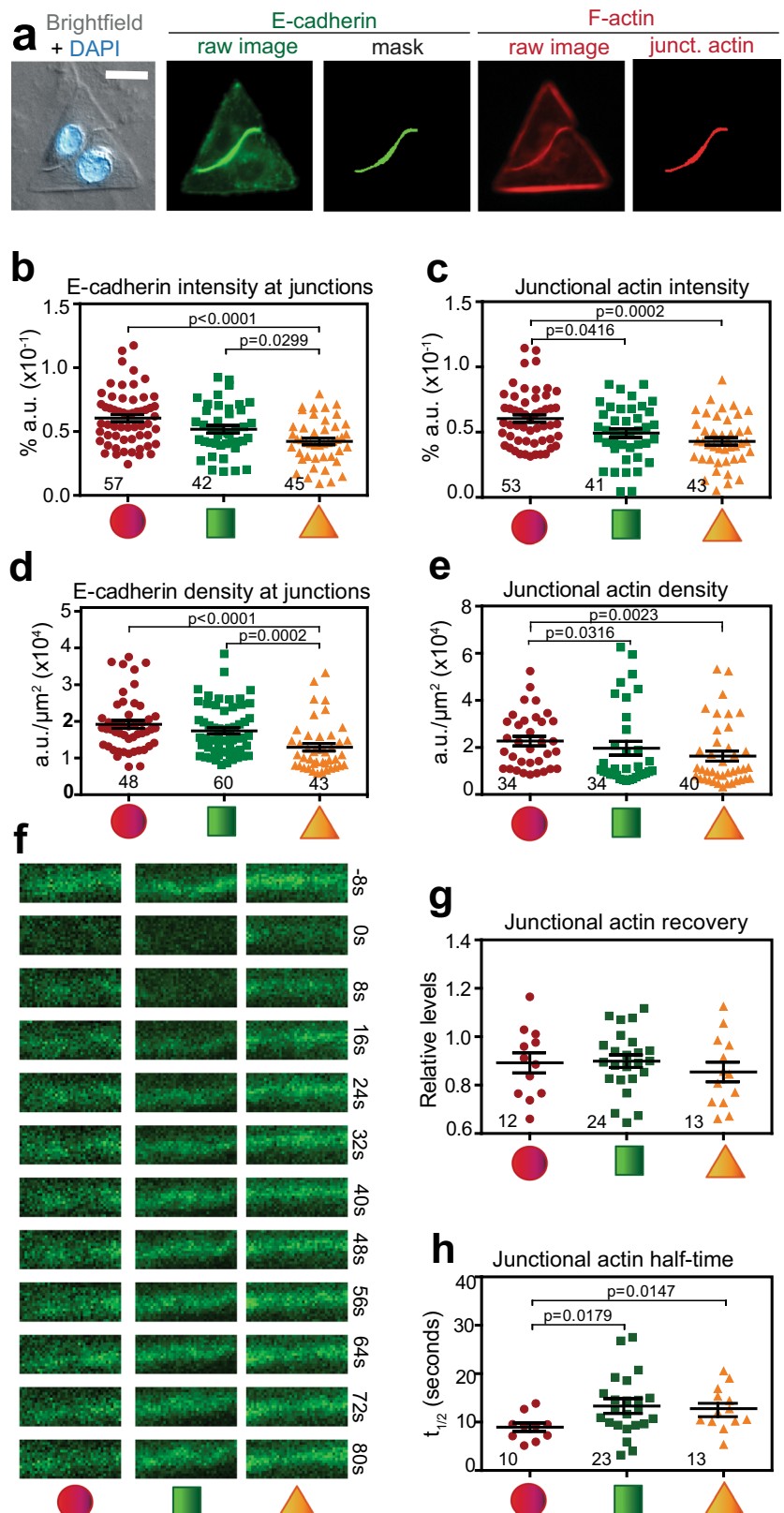

phenotype, i.e., undulation of cell–cell contacts and altered levels of E-cadherin and junctional actin. In keratinocyte monolayers, upon contraction inhibition thin bundles disappear, while junctional actin structures remain reasonably intact[51]. Following Y27632 treatment of confined cells, the density of junctional actin was not affected, irrespective of the geometric shape (Fig. 8a), suggesting that this F-actin pool is insensitive to relaxation. Although unexpected, the latter is

consistent with the low levels of contractility markers at junctional actin of keratinocytes on micropatterns (Fig. 4e, f) or monolayers[51,52]. In contrast, the density of E-cadherin at cell–cell contacts increased with contraction inhibition of cell pairs on squares and triangles, but not on circular doublets (Fig. 8b). Upon relaxation, junctions became significantly more curvilinear in squared and triangular-shaped cell pairs (higher linearity index, Fig. 8c). The junctions of treated triangular cell

**Fig. 6 | Highest tensile cell pairs have reduced density and intensity of E-cadherin and F-actin at junctions. a–e** Keratinocytes grown on different micropatterns were fixed and stained for E-cadherin, F-actin and nuclei. **a** Representative image showing the quantification pipeline: bright-field and wide-field images were processed to segment E-cadherin at junctions (mask) and then segment the corresponding area in the E-cadherin or F-actin images (junctional actin). **b–e** The intensity (% area) and the density (the ratio of intensity over segmented area of the marker at junctions) of E-cadherin receptors (**b**, **d**) or junctional actin (**c**, **e**) were quantified. **f–h** Actin dynamics at junctions were measured by FRAP. Representative video stills of GFP-actin are shown for each geometric confinement (**f**). Data were quantified to assess the maximum recovery obtained (**g**) and the amount of time necessary to recover 50% of GFP-actin fluorescence at junctions ($t_{1/2}$, **h**). Below graphs, micropattern geometries are represented as red circles, green squares and orange triangles. Number of junctions quantified is shown on top of each geometric shape ($N = 3$, **b**–**e**; $N = 6$, **f**–**h**). Scale bar = 20 μm. Mean values and error bars (SEM) are shown. Statistical analyses were done using one-way ANOVA, followed by Kruskal–Wallis post-hoc test and Benjamin, Krieger and Yekutieli test (**a**–**c**) or unpaired, two-tailed Mann–Whitney $t$-tests (**g**, **h**). Source data are provided as a Source Data file.

pairs contained cadherin receptors as brighter clusters and intensity (Fig. 8b), that however, covered less of the available contacting interface length (Supplementary Fig. 6e).

Our model fitting of the pharmacological relaxation data resulted in a very similar pattern of junction linearity (Fig. 8c, d), suggesting that the model fitted well with the experimental observations. In addition to the collapse of nuclei at junctions (Supplementary Fig. 6b, c), we assumed that treatment with Y27632 would modify primarily the combined active stress ($f_1$, $f_2$) of each cell, which considers cortical thickness, membrane stiffness and F-actin intensity (Fig. 4). The experimental E-cadherin density of Y27632-treated cells was used as input to determine the novel values of $\alpha_3$ and $\alpha_4$, coefficients that modulate the surface parameters in each neighbouring cell and keeping $\alpha_2$ and $\alpha_5$ unchanged. The model fitting indicated that Y27632 treatment modifies $\alpha_3$ and $\alpha_4$: (i) in opposite ways (Supplementary Fig. 6f), as seen with the distinct stiffness of neighbour cells on a given micropattern (Fig. 3d); and (ii) decreasing the sum of $\alpha_3$ and $\alpha_4$ following treatment with Y27632 (Supplementary Fig. 6f). This suggests that, even if the F-actin levels at junctions remains unchanged after drug treatment (i.e., as shown in Fig. 8a), the cortical stress of the cell pair must be overall reduced. We concluded that reducing mechanical stress alters the junction properties of a triangular cell pair to the phenotype seen on circular shapes: curvilinear contacts and, counterintuitively, higher density of E-cadherin receptor at junctions.

Our data indicate that circular-shaped cell pairs cannot generate sufficient intrinsic forces to shape up their junctions or alter receptor concentration consistent with the requirements of a mature cell–cell contact. We asked whether increasing the contractile properties of circular doublets is sufficient to improve the extent of junction maturation as evaluated by their configuration and levels of junctional markers. To increase cellular tension, we kept the area of attachment constant but varied substrate stiffness, thereby avoiding cell stretching. Normal keratinocytes were grown on circular shapes of different stiffness and the levels of F-actin and desmoplakin at junctions were quantified (as a proxy for desmosomal cadherins; Fig. 8e–g). On stiffer substratum, circular cell doublets straightened up their contacts in a more linear configuration and reduced the density of desmoplakin (Fig. 8f, g). In addition, on stiffer substratum, nuclei were further apart from each other (Fig. 8h), suggesting that additional mechanical forces contributed to nuclei repositioning. Thus, enhancing the intrinsic stiffness of circular doublets is sufficient to drive junction maturation as characterized by lower cluster density of junction markers, a straight junction and nucleus positioning at the cell centre.

## Discussion

Here, we demonstrate that cells modulate their intrinsic rheological properties to drive junction maturation, using adaptive mechano-responses to re-shape junction configuration and reorganize complexes at cell–cell contacts. A cell autonomous process to increase intracellular viscoelastic pressure or cortical stiffness (without stretching or external forces) is sufficient to promote junction maturation as a straight, taught cell–cell contact. Using a Junction Unit model (Fig. 9), snapshots of a dynamic interplay between cell doublets reflect the most frequent and preferred behaviour of cell pairs on each geometric shape. High-resolution global elasticity profiles show compliance variability at distinct nanostructures at the cell cortex and a progressive increase in cortical stiffness at cell apex, vertices, and junctions in triangular shapes. Within the range of intrinsic Young's modulus measured, cell pairs can form junctions and remain attached. Yet, the configuration and junction properties under various rheological status are remarkably distinct.

We find that cell rheology is responsive to neighbour viscoelasticity and environmental landmarks such as confinement geometry and asymmetric stress points. Consistent with a tug-of-war between two neighbours, cells sharing a micropattern have significantly different volumes, Young's moduli, and area, which mimics cell properties inside epithelial sheets. Volume fluctuations have been reported during epithelial morphogenesis[53], collective cell migration of epithelial monolayers[54], and in response to strain[49,55]. Although cortical stress variations in tissues have not been fully explored, fluctuations in junction vertex positioning, cell geometry and area accompany tissue folding, elongation and branching during morphogenesis[7,56]. Unexpectedly, junctions are the region with the lowest cortical stiffness in each geometric shape relative to cell apex or vertices. While counter-intuitive, our finding is consistent with dissipation of energy at junctions[57] to allow sustaining larger stresses elsewhere[49,55]. Lower tension at junctions is also observed following stretching[38,39], reduced traction forces on the substratum after monolayers are formed[58], or in regions where junctions are prominent[28,59] and cells motionless[60]. Thus, an emerging theme is the non-linear adaptive mechano-responses at junctions with distinct mechanisms of rebalancing stresses.

The unevenness of intrinsic stiffness and viscoelasticity of each neighbour has a major impact on the ability to attach to each other. Weaker, immature contacts with a curvilinear configuration are favoured in cell pairs with lower stiffness properties (Fig. 9a, b)[61,62]. In contrast, under conditions of higher intrinsic stiffness in triangular doublets, stable junctions are favoured as straight junctions flanked by parallel contractile thin bundles, similar to those in epithelial monolayers (Fig. 9)[15,63]. Straight junctions and parallel thin bundles appear to be the optimal configuration to resist orthogonal stresses at confined cell pairs[64,65] or tangential stresses in epithelial sheets, where multiple neighbours add extra directional forces and complexity[66,67].

We conclude that the cell autonomous modulation of intrinsic viscosity and stiffness is instrumental for the junction maturation process (Fig. 9). How localized deformation of the interface between neighbouring cells generate undulated junctions is presently unclear. Our modelling using a single junction informs distinct mechanisms to switch between undulated and straight junctions that considers three predictions. The first prediction is that curvilinear junctions appear because of unequal biomechanical properties between cell pairs, i.e., localized pressure at the contacting interface that can deform or bend cell–cell interfaces in various ways (Fig. 4). The model also predicts that the bulky nuclei of cell pairs have an impact on membrane

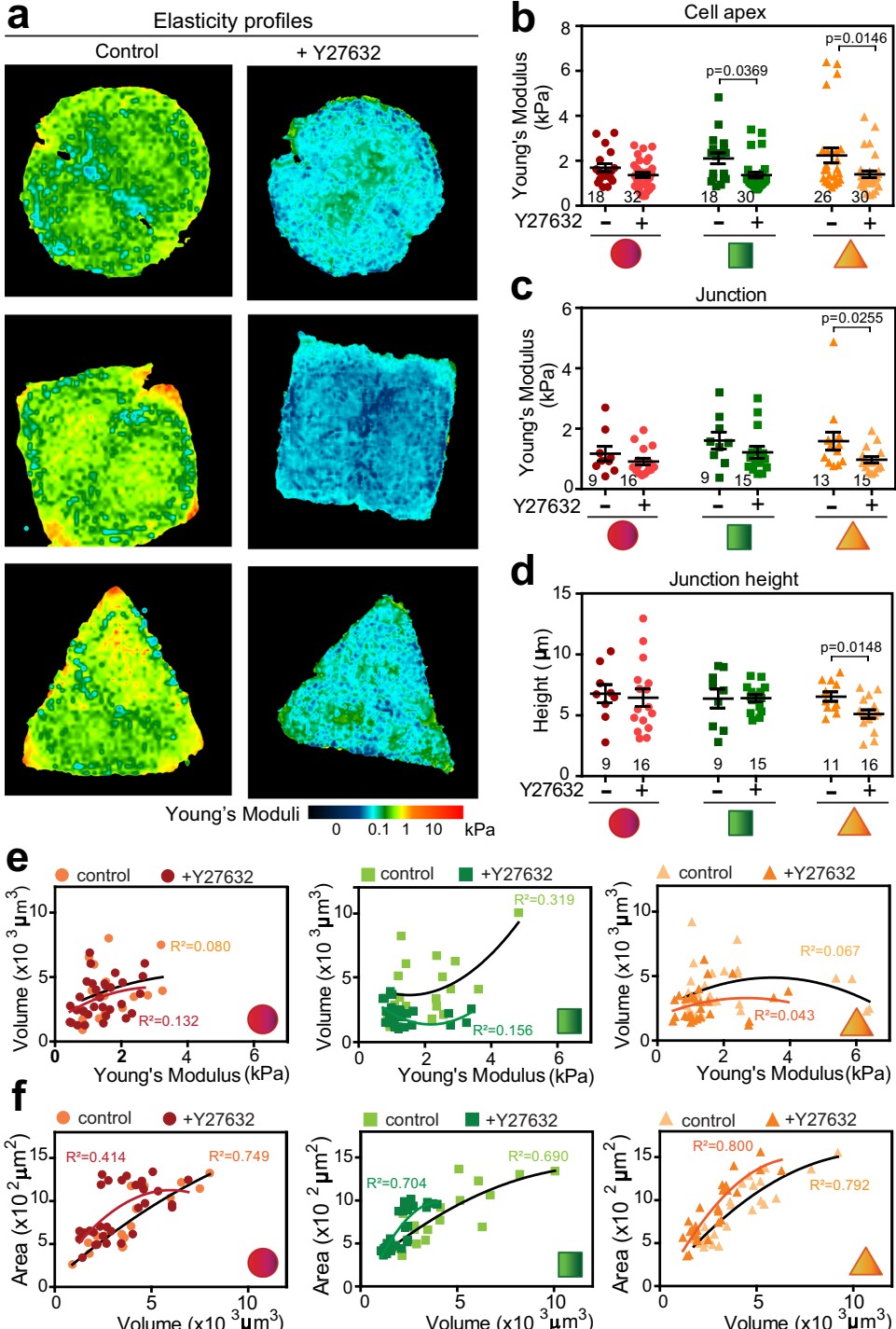

**Fig. 7 | Inhibition of intracellular contraction interferes with rheological properties of highest tensile triangular cell pairs, but not those on circular confinement.** Keratinocytes grown on different geometries were scanned using SICM in the presence or absence of Y27632. **a** Representative images of Young's moduli maps are shown for each sample. Colour map follows the scale at the bottom of images from softer (dark blue) to stiffer values (red). **b** and **c** Young's moduli values derived from (**a**) were plotted at the cell apex (**b**) or at junctions (**c**) of cell pairs on different geometries. **d** Height of the junction of confined cell pairs was measured in the presence of absence of Y27632. **e** and **f** Correlation between volume of an individual cell and its respective Young's moduli at the apex (**e**) or area

(**f**) is shown. Best fit curve was fitted and $R^2$ values calculated. Below and inside graphs, micropattern geometries are represented as red circles, green squares and orange triangles. Scan sizes shown in **a**: circles left 47 μm × 47 μm and right 44 μm × 44 μm; squares left 46 μm × 46 μm and right 38 μm × 38 μm; triangles left 47 μm × 47 μm and right 49 μm × 49 μm, Number of samples is shown inside graphs (**b**–**d**). $N = 7$. Mean values and error bars (SEM) are shown. Statistical analysis was done by Mann−Whitney test (unpaired, two tailed) to compare between untreated cells and those treated with Y27632 on each geometric shape (**b**–**d**). Source data are provided as a Source Data file.

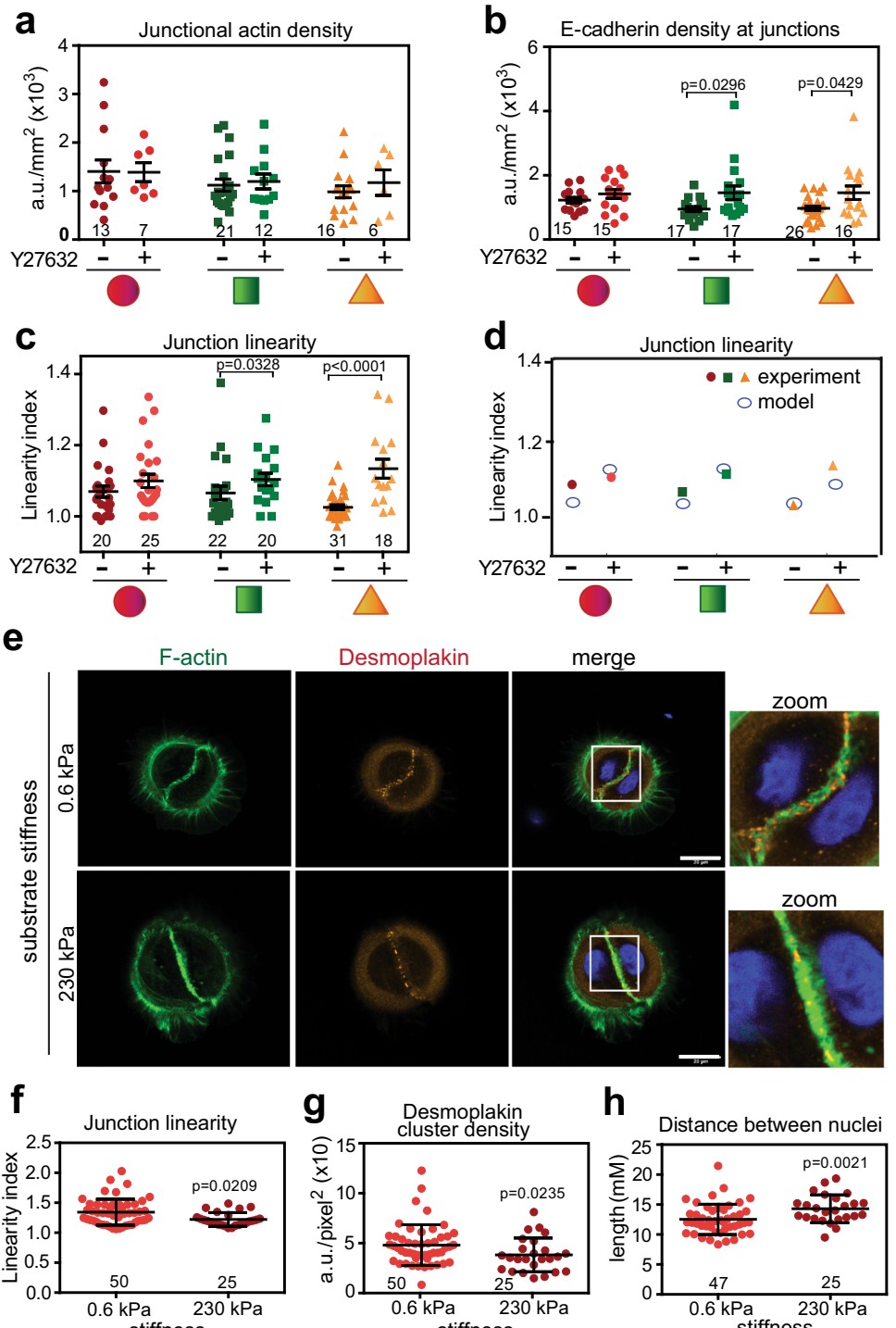

**Fig. 8 | Manipulation of intrinsic intracellular rheology is sufficient to drive junction maturation phenotype. a–c** Confined cell pairs incubated in the presence or absence of Y27632 were stained for E-cadherin or F-actin and images quantified to obtain density (**a, b**) or junction linearity (**c**) parameters. **d** Model fitting shows values obtained for junction linearity from experimental data (solid shapes) and computational modelling (open circles). **e–h** Keratinocytes were seeded onto various substrate stiffness on circular micropatterns with identical areas. Cells were fixed, stained and distinct parameters quantified. **e** Representative images of circular pairs stained for F-actin (green) and desmoplakin (red) as a readout for desmosomal cadherins. Merged images and zoom of the white square area are shown on the far-right columns. Scale bar = 20 µm. Quantification of how straight junctions are (**f**), desmoplakin cluster density (**g**) and the distance between cell pair nuclei (**h**) are shown for cells on different substratum stiffness. Micropattern geometries are represented below graphs as red circles, green squares and orange triangles. Number of samples quantified is shown on top of the treatment performed ($N = 3$, **a–c**; **f–h**). Mean values and error bars (SEM) are shown. Statistical analysis was done by one-way ANOVA, followed by Kruskal–Wallis post-hoc with Benjamin, Krieger and Yekutieli multiple comparisons tests (**a–c**) or unpaired, two-tails Mann–Whitney tests (**f–h**). Source data are provided as a Source Data file.

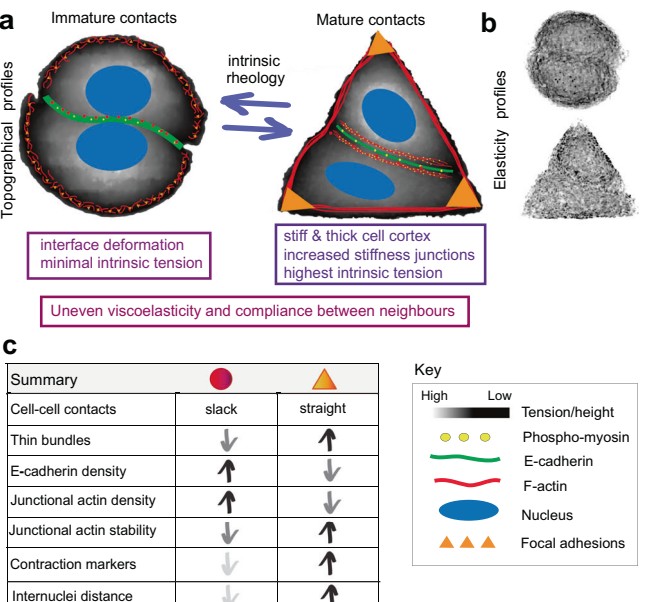

**Fig. 9 | Intrinsic cell rheology drives junction maturation—summary. a** Diagram depicting the topographical maps of circular and triangular cell pairs in grey scale, representing height as shorter (dark grey) versus higher length values (white) of cell pairs. Representation of junctions under different tensile stresses are shown as the shape of cell-cell contacts (E-cadherin, green), organization of actin filaments (red) at junctions and periphery of micropatterns as: lamellae (circles) or stress fibres (triangles), focal adhesions (orange colour) and labelling of PMLC (yellow dots). **b** Elasticity profiles of circular and triangular cell pairs highlight the presence of stiffer spots on regions/subcellular structures and junctions (white colour, hight stiffness) and areas of lower mechanical stress (dark grey). **a**, **c** Text boxes below diagrams (**a**) and summary table (**c**) show the identified rheological and molecular properties, respectively. In the table in **c**, micropattern geometries are represented as red circles or orange triangles.

deformation, particularly in circular cell pairs where nuclei collapse at either size of the contacting interface. Thus, the spatial integration of intracellular pressure, cell volume, nucleus positioning and cortical stress of a cell pair contributes to junction configuration. Future studies will highlight how expansion and retraction at contacting interfaces is coordinated by constrained variations of biomechanical stresses from neighbouring cells[68]. In addition to the internal pressure by the nucleus, it will be particularly interesting to dissect how the heterogeneous spatial distribution of receptors, channels, cytoskeleton and signalling along junctions contribute to their deformability as wavy or undulated shape.

The second prediction is that, to straighten up a curvilinear junction, stiffness and thickness of the cell cortex may be increased. The modulation of cell cortex thickness may result from a feedback mechanical response on a contractile F-actin meshwork to rebalance intracellular stresses[69–71]. While the precise mechanisms of cell cortex thickness in epithelia are unknown, it is likely that distinct mechanical adaptation are generated at different intracellular sites, depending on the force direction and amplitude, actin meshwork organization and density[72,73]. Experimentally, artificially increasing intracellular stiffness of circular cell pairs by attachment on stiffer substratum drives the appearance of a straight, taught cell–cell contact. Conversely, by reducing the stiffness of triangular cell doublets pharmacologically, the mature junction phenotype is replaced by a more compliant and curvilinear junction. We concluded that increasing intrinsic stresses is sufficient to trigger the transition from an immature to mature junction.

The third prediction of the model is that, under conditions of unequal cortical Young's modulus in each neighbour, cell–cell contacts can be straightened out by enhancing junction stiffness. Higher junction stiffness increases stress resistance and can be generated by two means: availability of adhesive receptors associated with cell–cell contacts or remodelling of the underlying F-actin network at junctions (see below). A positive correlation between cadherin levels at junctions and mechanical stress has been demonstrated[23,65,74] by altering the molecular components (levels, types of molecules) or junctional distribution of receptors by stiffness-dependent cadherin accumulation[75,76].

However, such positive correlation between augmented force and E-cadherin levels at junctions may not be a universal mechanism for junction stabilization. Triangular cell pairs with stiffer and straighter junctions have significantly lower density of cadherin receptors when compared to immature contacts of circular pairs (this work). Consistent with our findings, E-cadherin levels at junctions remain unchanged or reduced upon increasing forces with larger confined area[38], or following shear or compressive stress[77,78]. On the other hand, junctions disrupted by specific oncogenes[79] or depletion of specific actin binding proteins[80] have higher intensity levels of cadherin receptors at cell–cell contacts (albeit with different distribution and organization). Overall, the data indicate that junctional E-cadherin levels per se may not be a strong indicator of junction maturation or stable, reinforced contacts.

These apparently contradictory reports on receptor levels and junction responsiveness to mechanical stress remain to be conciliated. In some examples, it is likely that differential cellular responses result from a short-term mechanical stress (acute recruitment of receptors) versus a sustained mechanical stimulation, where adaptation responses are resolved as long-term maintenance of stable contacts[22,38,75,76]. It is also feasible that the type of stress and direction of forces may drive distinct feedback responses at the adhesive sites to counteract junctional tension[18]. Thus, distinct energy-dependent mechanisms of reinforcement may operate during junction maturation versus remodelling of pre-established contacts.

What is clear is that, at least in some conditions, increasing the effective concentration of E-cadherin at contacts is not an essential step for promoting resistance to increased stiffness (this work[2,81],), consistent with a force-dependent modulation of cadherin clusters at junctions[82]. Of note is that the disconnect between higher protein

levels at junctions and biomechanical stresses has also been reported in other adhesive systems such as desmosomes and tight junctions (ref. 18 and references therein). We conclude that an efficient immobilization of reduced amounts of cadherin receptors with a less dynamic F-actin network is sufficient to counterbalance higher stresses at junctions and cell cortices. As contractility markers are not enriched at junctional actin, alternative reinforcement mechanisms of the F-actin network must be in place to resist mechanical and viscoelastic stresses[83,84].

Understanding the detailed organization of the junctional actin pool in mature junctions will highlight novel principles of force-dependent adhesion stabilization. In vitro actin contractility studies[85] show that network architecture and connectivity by crosslinkers influence the fluidity and viscoelastic properties of shorter, looser filaments that buckle under pressure[71,83,86,87]. Indeed, actin crosslinkers and proteins controlling filament length (polymerization, capping, severing) emerge as important regulators of cadherin stabilization in response to stiffness, flow or assembly[80,88–90]. The known force-dependent interaction between cadherin complexes and F-actin[24] may also trigger recruitment of specific cytoskeletal proteins[88,91–93] that are likely to alter the junctional actin network organization[91,94].

Our study reveals an instructive role of intrinsic cellular rheology in the formation of a straight junction in epithelia and the unappreciated participation of biophysical and viscoelastic properties of cell neighbours in this process. In its simplicity, the global elasticity maps and the precise control of intracellular rheology entailed by the Junction Unit model are instrumental to unravel unforeseen mechanisms shaping up junction maturation and tissue architecture. The molecular mechanisms via which junctions respond to mechanical stress and mature remains an exciting topic for future exploration. Further studies will combine increased levels of complexity at multicellular scales and consider the impact of additional neighbours on the rheology-dependent junction maturation and functionality. Multicellular complexity will provide a fascinating springboard to dissect the interplay between intrinsic rheology, junction maturation and plasticity in various physiological, regenerative, and pathological events.

## Methods
### Microcontact printing
Micropatterned substrates with fibronectin coating were fabricated using a modified protocol as described[95,96]. A silicon wafer master prepared by photolithography (gift from M. Textor, ETH Zurich) was used to make elastomeric stamps through replica casting polydimethylsiloxane (PDMS, Dow Corning) cured overnight at 60 °C. For protein adsorption, the pattern containing surface of the stamp was fully coated with fibronectin (Sigma) at a concentration of 50 µg/ml for 1 h. Coated stamps were blown dry and placed in conformational contact with the non-treated substrate surface for 1 min before peeling off. Substrates were then immersed in 0.5% (w/v) Pluronic-F27 (Sigma) for 1 h to passivate non-adhesive areas.

Alternatively, surface micropatterning was performed using a PRIMO system (Alvéole Lab, France) according to the manufacturer's instructions. Circular micro-patterns (diameter = 48 µm, growth area = 1800 µm²) were designed in Inkscape (http://www.inkcsape.org/) as 16-bit binary file and exported as TIFF file via FIJI to upload into the Leonardo software v4.12 (Alvéole Lab, France). In detail, elastomeric substrates were incubated with 750 µl of a 0.01% poly-L-lysine solution (Sigma) at 4 °C overnight for surface passivation. The surface was washed three times with 1× PBS (pH 7.2) followed by washing three times with 100 mM HEPES buffer (pH 8.4) (Sigma). An antifouling layer was formed on a 2D shaker at RT for 1 h in dark environment using 500 µl methoxy-poly-ethylene-glycol-succinimidyl valerate (mPEG-SVA, LaysanBio, USA) of a 50 mg/ml in 100 mM HEPES stock solution. Reaction was inactivated by washing three

times with HEPES buffer and five times with 1× PBS. Subsequent micro-patterning was performed on a Nikon Eclipse Ti2 microscope equipped with a 355 nm pulse laser, PRIMO devices and a ×20/0.8, S Plan Fluor EWD objective. By using 40 µl of photoreactive PLPP (Alvéole Lab, France) about 180 circles were structured with a UV light exposure of 1200 mJ/mm² (40 s each DMD) and a laser setting of 5 V. Before use, surfaces were washed 5 times with 1× PBS and stored in the dark at 4 °C if not used directly.

For preparation of silicone rubber-coated cell culture substrates, Sylgard 184 elastomer kit was used to produce elastomeric substrates with 0.6 kPa (ratio of base oil to cross linker: 73:1) (w/w) and 230 kPa (ratio: 27:1) (w/w) stiffness. Preparation was performed as described before[97]. Before cross-linking, elastomer was spin-coated over cover glasses to form approximately 70 µm thick layers. Coated cover glasses were then glued directly to the bottom of predrilled cell culture dishes (total diameter Ø = 35 mm, cultivation area Ø = 18 mm). Cross-linking was performed at 60 °C for 16 h.

### Cell culture
Normal human keratinocytes from neonatal foreskin (strain Sf, passages 3–5) were cultured on a mitomycin C (Sigma)-treated monolayer of 3T3 fibroblasts at 37 °C and 5% CO₂ in standard medium containing 1.8 mM CaCl₂. Upon reaching 90% confluence, keratinocytes alone were seeded directly onto micropatterned substrates without the addition of fibroblasts and incubated overnight for optimal attachment and spreading. Mycoplasma tests are done routinely in the lab, and our cell stocks are mycoplasma-free.

For SICM experiments, live cells were scanned in FAD medium buffered with 25 mM HEPES pH 7.4. For elasticity dependent cell doublet analysis, elastomeric substrates were precoated with 500 µl fibronectin solution (0.1 mg/ml, Corning, USA) at 37 °C for 30 min. Subsequently, $2.0 \times 10^5$ neonatal normal human keratinocytes (CellSystem, Germany) were seeded per substrate and cultivated in DermaLife K calcium-free medium (CellSystem, Germany) without EGF to allow adhesion under cell culture conditions for 20 min. Unattached cells were removed by washing three times with DermaLife K. For further incubation medium was replaced by DermaLife K cell medium with 1.8 mM calcium to develop cell–cell contacts.

Treatment with Y27632 (Sigma) at a concentration of 5 µM was performed on micropatterned cells by pre-incubation for 5 min (staining micropatterns) or 60 min (SICM analysis). Transfections were carried out on cells seeded on micropatterns with FuGENE® HD reagent (Promega) using the pCDNA3.1.GFP-β-actin construct (gift from M. Bailly, University College London) and expressed overnight.

For FRAP experiments, cells were seeded onto tailor-made thin bottom dishes and maintained in standard calcium medium. Throughout imaging, cells were kept at 37 °C in phenol red-free DMEM/F-12 (1:1) medium with HEPES (Gibco, Life Technologies) supplemented as described for keratinocyte cultures. For SICM experiments, untreated dishes (NUNC) were used to seed cells that were maintained in standard calcium medium and buffered with HEPES.

### Immunofluorescence and microscopy
For immunostaining, keratinocytes were fixed in 3% paraformaldehyde for 10 min, permeabilized with 0.1% Triton and blocked with 10% FCS for 10 min. Staining was carried out as described in ref. 98 or as described in ref. 99 for doublets on elastomeric substrates Antibodies used for fluorescence were against E-cadherin at 1:1000 (HECD-1, gift from Prof Takeichi) or ECCD2 at 1:750 [#13-1900, Invitrogen]), anti-desmoplakin I antibody at 1:200 (guinea pig, #708251A, DP1, Progen) and myosin light chain phosphorylated at Ser19 at 1:1000 (#3675S, Cell Signalling). Secondary antibodies were bought from Jackson ImmunoResearch: Alexa Fluor 488 (AF488)-conjugated anti-mouse IgG (host goat, #115-545-003, 1:1000); Indocarbocyanine (Cy3)-conjugated

anti-mouse IgG (host donkey, #715-165-151, 1:3000); Indodicarbocyanine (Cy5)-conjugated anti-rat IgG (host donkey, #712-175-153, 1:400) and Indocarbocyanine (Cy3)-conjugated anti-guinea pig (host goat, #106-165-003, 1:200). F-actin labelling of was done using AlexaFluor-568 phalloidin (1:1000, Invitrogen) or Phalloidin Atto 488 (1:500, Sigma). DNA was labelled using DAPI (Sigma) or NucBlue® (Invitrogen).

Wide-field images were acquired on an inverted microscope (Zeiss Axio Observer) with EC Plan Apochromatic objectives (×20 numerical aperture (NA) 0.8 DIC and ×40, NA 0.75 Ph2). Confocal images were acquired on an inverted scanning confocal (Zeiss 780 LSM laser) using a DIC Plan Apochromatic objective (×63 NA 1.4 Oil). The software used in both cases was Zen software (Carl Zeiss). Alternatively, imaging of doublets seeded on elastomeric substrates used a confocal Laser Scanning Microscope 880 (cLSM 880, Carl Zeiss) equipped with a water immersion objective (C-Apochromat ×40/1.2 W Corr M27). All microscopy settings were kept identical throughout all experiments for best comparability. Images were taken close to the basal plane of the substrate surface with the focus on imaging cell-cell contacts.

Fluorescence recovery after photobleaching (FRAP) experiments were conducted on an inverted confocal microscope (LSM-510) with a DIC Plan Apochromatic objective (×63 NA 1.4 Oil). Five pre-bleach images were acquired before photobleaching of a 14 × 35 pixel region at a central part of cell–cell contacts using the 488 nm laser at 100% laser power with 50 iterations. Post-bleach imaging was performed at 4% 488 nm laser power for 192 s with 8 s intervals. Only junctions between expressing and non-expressing cells were imaged.

## Scanning ion conductance microscopy (SICM)

SICM is a non-contact microscopy technique based on the principle that the current (flow of ions) through the nanopipette filled with electrolytes decreases when the pipette reaches the surface of the sample[100,101]. It offers some advantages when compared to AFM[102–105], mainly as the use of a non-invasive probe, the quality of the images obtained and ability to measure elastic modulus as low as 10 Pa. All images in this study were recorded using SICM scanner from live cells using a variant of SICM called hopping probe ion conductance microscopy[40]. Images were observed and analysed using SICM image viewer. Two different setups were use in this work. For the hydrojet pressure-application experiments (Fig. 1 and Supplementary Fig. 2) the scan head (ICnano Scanner Controller, Ionscope Ltd) was controlled by the *xyz* piezo three-axis translation stage Triton-100 (Piezosystem, Germany) with 80-μm closed-loop travel range in *x*, *y* and *z* directions. The pipette electrode head-stage was connected to a Multiclamp 700B amplifier (Molecular Devices). The system was placed on the platform of a Nikon Eclipse Ti-E inverted microscope (Nikon Corporation, Japan). For the stiffness mapping experiments (Figs. 2, 3, 7 and Supplementary Fig. 3), the scan head (ICnano Scanner Controller, Ionscope Ltd) was controlled by a three-axis piezo-translation system (Physik Instrumente, UK) with a 100 μm × 100 μm *x*–*y* piezo-stage for sample positioning and 38 μm *z*-axis piezo-actuator for the vertical movement of the pipette. The pipette electrode head-stage was connected to an Axopatch 200A amplifier (Molecular Devices) mounted on the stage of a Nikon Diaphot 200 inverted microscope (Nikon Corporation, Japan). Nanopipettes (20–30 MΩ tip-resistance) were pulled from 1.0 mm O.D. 0.5 mm I.D. borosilicate capillary glass using a laser puller (P-2000, Sutter Inc.) and filled with buffer salt solution for all experiments.

For the measurement of displacement, recovery and relaxation a ramp pulse of air pressure was delivered by displacing air connected to the auxiliary inlet in the pipette holder, generating a hydrojet of the intracellular pipette solution; the delivery was controlled by an electric valve via Digidata 1440A (Molecular Devices, UK). Measures were derived from the nanopipette's vertical displacement (*Z*-direction), acquired using pClamp 10.0 (Molecular Devices). Relaxation time (time taken for cells to reach a stable baseline level post pressure application)

and recovery (percentage of cell that reached initial baseline after pressure application) were analyse using Clampfit 10.7 (Molecular devices) from the nanopipette's vertical displacement recordings.

To maximize the speed of scanning, sizes were adjusted as it does not affect the measurements. For the mapping of Young's modulus, cell and junction heights, we used the method described in detail by Rheinlaender and Schaffer (which assumes an elastic medium)[44] and with a software modification as reported by Swiatlowska and colleagues[41]. Briefly, a constant pressure of 15 kPa was applied to the auxiliary inlet while recording topography using the hopping mode[40] at two different set points (current drop of 1% and 2%) simultaneously. The difference between the heights of the two topography images represented the displacement *d* of cell membrane due to applied pressure $p_0$ Young's modulus *E* was then calculated as

$$E = p_0 A \left( \frac{d}{d_{substrate}} - 1 \right)^{-1}$$

where $d_{substrate}$ is displacement observed when imaging uncompressible substrate (bottom of the cell culture dish), and *A* is parameter reflecting the geometry of the pipette as defined by Rheinlaender and Schaffer[44]. To correct for the effect of local slope, a correction described in Fig. S4 was applied to the displacement data prior calculation of the Young's modulus. The sizes of images obtained for elasticity maps were 40 × 40 μm to 53 × 53 μm (Fig. 2) and 44 × 44 μm to 47 × 47 μm for samples treated with Y27632 (Fig. 7).

## Image processing and quantification

In-house macros were developed in Fiji (https://fiji.sc/) to measure nuclear distances, nuclear (distance between the centre mass of segmented DAPI-stained images) and cell areas (segmented individual cells on a micropattern using the E-cadherin image), Measurement of junction intensities were carried out, whereby the E-cadherin image was thresholded to seclude the junctional region and minimize the contribution of cytoplasmic staining. The thresholded E-cadherin image was then used as a mask on the corresponding phalloidin-stained image to segment the junctional actin pool. Intensity of E-cadherin or F-actin was calculated as the summed brightness of their respective segmented junctional region. Density of a marked at junctions was assessed by taking the ratio between its intensity divided by its segmented area. The in-house software Junction Mapper was also used for some experiments[79]. Peripheral structures and thin bundles flanking junctions were qualitatively assessed using confocal images. Young's modulus calculations were conducted as described above.

For junction positioning, cell pairs that showed junctions at a particular position were quantified. Junctions of circular doublets usually span the equator. Squared and triangular cell pairs were split into two categories considering whether or not the junction passed through a vertex of the shape. The number of cell pairs with junctions fulfilling each of the categories was then counted and expressed as a percentage of cell pairs. The likelihood of random orientation of junctions at 5° degree intervals was calculated by dividing by 72 for circular pairs (i.e., 360°/5°). For squares and triangles, the probability was calculated at each vertex (i.e., 20° for squares and 15° for triangles) or at the remaining 5° intervals at the sides of the shape.

During FRAP experiments, actin recovery was quantified using an ImageJ method adapted from[106]. Briefly, a region of 14 × 35 pixels was drawn over the bleached area and raw integrated intensity values was measured. ROI location in cells moving during the video recording was manually adjusted as necessary. To remove noise, mean intensity values of background was measured in a large area outside the cell, was normalized to the size of the bleached area and was subtracted from the raw integrated intensity values of the bleached area. To account for photo-bleaching, the corrected intensity values in the bleached area were further divided by mean cytoplasmic intensity as measured in a

large region covering the cytoplasm of the transfected cell. The resulting normalized intensity values were fitted with a single exponential function in GraphPad Prism, yielding recovery plateau and recovery half-time. Only curves with $R^2$ values above 0.7 and reaching a plateau during the observation period were included in the analysis.

### Statistical analysis

Graphs were created and statistical analysis performed using PRISM (GraphPad versions 5 (correlations) or 9) and Excel. Data generally did not show normal distribution and were analysed using *t*-tests, Mann–Whitney (unpaired, two-tailed), Wilcoxon matched pair test, Statistical analyses were done using one way ANOVA, followed by Kruskal–Wallis post-hoc with Benjamin, Krieger and Yekutieli multiple comparisons tests and two-way ANOVA, followed by Šidák's multiple comparison post-hoc test.

### Reporting summary

Further information on research design is available in the Nature Research Reporting Summary linked to this article.

## Data availability

The data generated in this study are provided in the Supplementary Information/Source Data file. Raw image files are deposited in Figshare (https://doi.org/10.6084/m9.figshare.c.6094764). Any further details and reagents are available from the corresponding author upon reasonable request. Source data are provided with this paper.

## Code availability

The programme Junction Mapper has been described elsewhere[79] and is available for download from https://dataman.bioinformatics.ic.ac.uk/junction_mapper/index.html or via the https://doi.org/10.5281/zenodo.6563424. Junction Mapper code is also deposited in GitHub https://github.com/ImperialCollegeLondon/Junction_Mapper. Custom-made code for analysing SICM images is available from https://github.com/PavelNo/SICMImageViewer.

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

## Acknowledgements

We would like to thank the initial optimization of SICM experiments by G. Haywood and microcontact printing experiments by Dr. T. von Erlach. We thank Prof. M. Textor (ETH, Switzerland) for the micropattern master and S. Liu (Johns Hopkins University) for help with the modelling. Work was supported by BBSRC (BB/M022617/1 to V.M.M.B.) and British Heart Foundation grants RE/08/002/23906 (V.M.M.B.) and RM/13/1/30157 and RG/12/18/30088 (J.G.). M.M.S. acknowledges support from a Wellcome Trust Senior Investigator Award (098411/Z/12/Z) and an ERC Con-solidator grant "Naturale CG" (agreement no. 616417). Deutsche For-schungsgesellschaft (DFG, German Research Foundation) supported work through (363055819/GRK2415) to R.M. and through SPP1782 within the projects HO2384/2 (B.H.) and ME1458/8 (R.M.). We thank F. Pichaud for comments on the manuscript and G. Dreissen for help with image processing. The authors also acknowledge the use of the Facility for Imaging by Light Microscopy (FILM) at Imperial College London.

## Author contributions

K.S.-R. conceived, designed, optimized, and performed experiments, J.L.S.A. and P.S. performed SICM experiments and analysis, S.R. helped with image analysis algorithms and P.N. provided the SICM algorithms. S.X.S., and S.L. performed the modelling. D.K., S.G., B.H. and R.M. per-formed the experiments on different substrate stiffness and helped with writing. J.G., M.M.S., and V.M.M.B. conceived, designed, and managed the project. K.S.-R., J.L.S.A., S.X.S., and V.M.M.B. wrote the manuscript.

## Competing interests

The authors declare no competing interests.
