## [Peer Review File · Nature Communications]

Intrinsic cell rheology drives junction maturationEditorial Note: Parts of this Peer Review File have been redacted as indicated to remove third-party material where no permission to publish could be obtained.

REVIEWER COMMENTS

Reviewer #1 (Remarks to the Author):

The paper by Sri-Ranjan et al entitled "intrinsic cell rheology drives junction maturation" endeavors to establish a relation between the morphology of a cell-cell contact and the rheological properties of the cells. To do so they use cell doublets plated on patterns of various sizes to control the tension/rheological properties of the cell based on cell spreading area. They cleverly exploit the asymmetry of the two cells spread area to measure asymmetric distribution of rheological properties and their consequences on the junction morphologies. They use ion scanning microscopy to infer the local rheological properties and profiles of the cells.

Their most robust observation is the fact that the cells have an unstable and slack junction when plated in circular patterns while they develop straighter junctions in triangular or square patterns. They correlate/attribute these observations to the rheological properties of the cells measured with the ion scanning microscopy.

I found the idea of the paper interesting and timely in our understanding of how cell regulation of its intrinsic rheological properties impact/are impacted by cell-cell contact. I fully appreciate that try to establish a link between the two process is experimentally hard. I also believe that the authors have a good system and an interesting approach. However, it was quite difficult for me to understand clearly what the authors are precisely trying to measure and correlate.

I hence have a few major concerns:

- My main difficulty with the paper lies in the fact that the authors use a terminology to describe the rheological notions in a quite confusing way. They seem to use stiffness, tension, elasticity, visco-elasticity, elastic modulus, even compliance in an almost interchangeable way. It would be much easier to understand what the authors really mean if they only try to describe their experiments in terms of intrinsic properties of the cytoskeleton (Young modulus), or only cortical tension and avoid the term stiffness, viscoelasticity, compliance.... after reading several times the paper I am still unclear what rheological properties the (cytoskeleton, cortex, nucleus...) they claim to measure. To me they measure a mixture of that can certainly boil down an effective modulus. But this should be clarified. I hence suggest that the authors rewrite the paper with a clear claim on what rheological properties they are talking about.

- My second concern is about the model. It would require a much longer description in supplementary doc for eg. The authors should discuss the validity of their hypothesis. They should also much more elaborate on why they can obtain the sinusoidal like interface or even the slack interface with equal pressure on each side. Assuming a gradient of tension along the junction means that there must be a flow of material along the contact. Is the local mechanical equilibrium broken in the model? Are the predicted junction shapes transient or calculated at equilibrium? I could not find this information in the paper. The way the introduce the parameters a_1 to a_6 is also quite problematic: they should have dimensions, but they are treated as dimensionless parameters. Additionally, the authors do not discuss (unless I missed a supplementary text I could not find) the value they find in their fit and whether these values make physical sense. This is the only way to assess if the model has a chance to capture the underlying mechanism. Hence it is impossible for me to judge if the explanations provided by the author make quantitative sense or not. I would like to be convinced that the elasticity of the cadherin molecules k plays a critical role in the story because it is not the first thing I would have guessed.

For all these reasons it is difficult for me to judge the quality of the paper. The experiments are sound, and the analysis is ok. I do not ask for more data, but clearly the interpretation, the explanation and the modelling have to be improved to judge of the scientific impact and interest of the work.

Minor comments:

- It would have been easier to number each line
- The parameters of the model are randomly labeled alpha or a, including in Figure S5.

- Works from like :Liu, Z. et al. Mechanical tugging force regulates the size of cell–cell junctions. Proc.Natl Acad. Sci. USA 107, 9944–9949 (2010).,Maruthamuthu, V., Sabass, B., Schwarz, U. S. & Gardel, M. L. Cell–ECM traction force modulates endogenous tension at cell–cell contacts. Proc. Natl Acad. Sci. USA108, 4708–4713 (2011)., Tzeng et al PNAS January 31, 2012 109 (5) 15061511;https://doi.org/10.1073/pnas.1106377109 have already described the interplay between cell shape and junction geometries and could be cited.
- The sentence ending by “the product of number density and stiffness of the cadherin bonds” is unclear
- Figure 4H should zoom on the results rather than have a large span on the y axis
- In my compiled version the letters identifying each sub panel are gigantic. They could certainly be reduced to ease the reading of the figure.
- When the authors say: Having shown the distinct F-actin organization in the cytoplasm of more tensile neighbours, we inferred that higher stiffness of triangular cell pairs (Fig. 2A-B) would drive increased levels of E-cadherin at junctions to enable cells to sustain the increased mechanical load at cell-cell contacts. “Do they consider the tangential stress or the component of the stress orthogonal to the junction?”
- In the sentence: “similar pattern of junction linearity with and with contractility inhibition” a word is missing.
- I do not quite understand what this sentence means.” Here, we demonstrate that cells modulate their intrinsic rheological properties and drive junction maturation by reorganizing the configuration and molecular organization of cell-cell contacts.”
- In the sentence: “A cell autonomous process to increase intracellular pressure and cortical stiffness (without stretching or external forces...” I would have changed and in or
- Why do you introduce the notion of viscoelasticity at the end of the paper when you never mentioned anything about viscous dissipation and treated the prob as a purely elastic problem?
- To derive the equation to measure the young modulus using SICM, do you assume that the medium is elastic? What Young modulus do you measure? That of the cortex? It seems likely that it is more the bending modulus that is measured that depends on the cortical tension. Could you elaborate her a bit to what part of the cell this modulus is associated to?

In a nutshell, the paper is promising, the experimental data are ok but the interpretation and the modelling as to be clarified since they constitute the main interest of the paper I believe.

Reviewer #2 (Remarks to the Author):

This paper describe interesting experiments aimed at gaining a deeper understanding of the elasticity and geometry of cell-cell junctions in well-controlled geometry of a cell pair. The results are interesting, and uncover non-trivial relations between the geometry and the junction's shape and cell elasticity, and the local organization of the actin, myosin and cadherin adhesion bonds.

I would recommend publication, if these comments can be addressed:

1) This is relevant citation with respect to the fluctuations in the monolayer:

Fodor, É., Mehandia, V., Comelles, J., Thiagarajan, R., Gov, N. S., Visco, P., ... & Riveline, D. (2018). Spatial fluctuations at vertices of epithelial layers: quantification of regulation by rho pathway. Biophysical journal, 114(4), 939-946.

2) By speaking about "viscoelasticity" of the junction they give the impression its a passive material property, while it relies on ATP-consuming processes related to acto-myosin contractility and actin polymerization. This should be explained.

3) I have a general problem: isnt a 2-cell system inherently less stable, unlike a 3-cell system, which has a very stable tri-cellular junction ? wouldn't it be more revealing and relevant for tissues and monolayers to study a 3-cell system ?

4) page 5, English typos: "ability of attach to each other", "each neighbouring interacts", page 6: "cadherin bondsWe assume".

More typos at the top of page 8.

5) Does the local Young's modulus correlate with the local fluorescence intensity of actin, or myosin, or actin+myosin ? can they show such a comparison.

6) Where in equations (1) do we see that if the junction is longer it can allow for more adhesion sites and therefore provides a larger negative adhesion energy that can be a driver to increase the length ? I don't see this adhesion-derived term in these equations.

7) On page 9 they write: "Yet, inhibition of intracellular stiffness did not alter the strong correlation between area and volume of each cell". Why should it ?

8) They write: "In contrast, the density of E-cadherin at cell-cell contacts increased with contraction inhibition of cell pairs on squares and triangles, but not on circular doublets (Fig. 8B). Upon relaxation, junctions became significantly more curvilinear in triangular-shaped cell pairs (higher linearity index, Fig. 8C)." Isn't this a possible indication that: The increase in cadherin at the junction on the triangular shapes may drive the increase in junction length, to allow for larger "wetting" of the two cells.

9) Are the cells on the circular islands rotating ? this could also affect the shape of the junction.

10) What is so special about shapes with sharp corners ? they seem to be sites from which cell-spanning stress-fibers emanate. Are these indeed the main drivers of the differences ? How does the cell-cell junction change when the SF at the sharp corner are laser-ablated, for example ?

11) In Fig.1B I think that in the bottom image the text is exchanged.

12) Fig.2A: In this panel the circular cells seem stiffer than those on the square, right ? this is against the reported trend.

13) Fig.3: Give the distribution of the ratio between the two cells for each geometry, to clearly see how the geometry affects the asymmetry between the neighbors.

14) Fig.3E: Seems that the junction position always chooses approximately the longest straight path: equator in the circle, diagonal in the square and the altitude in the triangle. Is this to maximize the contact area and therefore maximize the adhesion energy ?

Reviewer #3 (Remarks to the Author):

In this manuscript, Sri-Ranjan et al. presented a study on junction maturation in cell-cell contact with scanning ion conductance microscopy (SICM) and fluorescence microscopy. The new models of intrinsic cell rheology modulation offer a more complete understanding of cell junction maturation. The authors implemented an improved SICM to image the topography and extract mechanical properties of neighboring cells in different configurations. The quality of the SICM data is good. The paper is well-written, and its data supports the conclusion. However, I have some concerns regarding the use of fixed cells in this study. The authors used a cross-linking fixation agent that may change the mechanical properties of the cells and therefore the findings and conclusion presented in this manuscript may be compromised. I believe the authors should address this issue to enhance the acceptance of these findings by the scientific community.

1. SICM is a unique tool that offers high-resolution topography and stiffness measurements of the soft eukaryotic cell membrane without mechanical contact. I'm glad the authors used SICM instead of AFM, which is a more established technique but may not be able to detect lower stiffness regions. What's the sensitivity of the stiffness measurement with the SICM method used in this study? The authors should consider performing some AFM experiments to compare and validate the SICM stiffness measurements?

2. A major concern is the use of fixed cells in this study. Paraformaldehyde (PFA) crosslinks the many molecules and structures in the cell. Therefore I would expect an increased stiffness of the measurements compared with live cells. It could be that the use of fixed cells still shows similar relative behaviors, but the absolute numbers are almost certainly wrong. For this manuscript to be acceptable, the authors would need to either repeat the measurements with live cells, or at the least show comparison measurements between live and fixed cells. Depending on these results, perhaps the authors can explain why using fixed cells is acceptable.

3. Why did the authors use PFA 3% fixation? Was it the one they expect to affect the measurement the least? If so, why? If the authors continue the path with the fixed cells, I propose they test several fixation conditions. PFA 4%, 2%, w/o triton. Triton may also change the mechanical properties measured as small holes perfuse and degrade the membrane. The authors should show the effect of the fixation on the measurements.

REBUTTAL - REVIEWER COMMENTS

Reviewer #1

We thank the reviewer for his/her opinion that our paper is “interesting and timely in our understanding of how cell regulation of its intrinsic rheological properties impact/are impacted by cell-cell contact”. The reviewer appreciates the experimental challenges to correlate cell rheology with cell biochemical properties and values our unique system and interesting approach. The reviewer asks for more explanations on our interpretation and the modelling to improve the scientific impact and interest of the work.

1 - My main difficulty with the paper lies in the fact that the authors use a terminology to describe the rheological notions in a quite confusing way. They seem to use stiffness, tension, elasticity, viscoelasticity, elastic modulus, even compliance in an almost interchangeable way. It would be much easier to understand what the authors really mean if they only try to describe their experiments in terms of intrinsic properties of the cytoskeleton (Young modulus), or only cortical tension and avoid the term stiffness, viscoelasticity, compliance.... after reading several times the paper I am still unclear what rheological properties the (cytoskeleton, cortex, nucleus...) they claim to measure. To me they measure a mixture of that can certainly boil down an effective modulus. But this should be clarified. I hence suggest that the authors rewrite the paper with a clear claim on what rheological properties they are talking about. We thank the reviewer and have revised the whole text to improve clarity. This is shown as track changes in the submitted revised manuscript.

2 - My second concern is about the model. It would require a much longer description in supplementary doc for eg. The authors should discuss the validity of their hypothesis. They should also much more elaborate on why they can obtain the sinusoidal like interface or even the slack interface with equal pressure on each side. Assuming a gradient of tension along the junction means that there must be a flow of material along the contact. Is the local mechanical equilibrium broken in the model? Are the predicted junction shapes transient or calculated at equilibrium? I could not find this information in the paper. We thank the reviewer for pointing out the unclear points. We now provide a Supplementary Note and a new Supplementary Fig. 5 elaborating on our modelling. We also re-wrote the section of the manuscript describing the model, with added information and clarification of the parameters used (pages 6-8). We have additionally included the points that:

- junction shape is calculated as informed by the static images obtained experimentally.
- the junction shape calculation is a mechanical equilibrium calculation using force balance in the normal component.
- when equal pressure on each neighbour is present (Fig. 4D,G), the slack shapes can also be driven by unequal cortical stress (parameter σ). The implication of such scenario is that cells have to readjust their volume or area accordingly. Cell-cell contacts can be straightened up by altering junction stiffness or re-equilibrating cortical stress among neighbours. Slack junction shape would suggest a dynamic, more localized fluctuation of pressures along contacts, coordinated by both neighbours.
- the sinusoidal-like interface can be generated by unequal hydraulic pressure along the contacting interface. The reviewer is correct that it implies that a flow of material should occur along the contact. While our model does not consider tangential forces, previous evidence indicate that a flow of material is precisely what happens in a dynamic status such as collective migration.¹
- in addition, the sinusoidal shape of the junctions can be influenced by nuclei positioning juxtaposed to cell-cell contacts, which is particularly prominent in circular cell pairs or in drug-treated squared and triangular doublets (e.g., Supplementary Fig. 6B,C).

The way the introduce the parameters a_1 to a_6 is also quite problematic: they should have dimensions, but they are treated as dimensionless parameters. Additionally, the authors do not discuss (unless I missed a supplementary text I could not find) the value they find in their fit and

whether these values make physical sense. This is the only way to assess if the model has a chance to capture the underlying mechanism. Hence it is impossible for me to judge if the explanations provided by the author make quantitative sense or not. We added additional information on the main text (page 8), a new supplementary information (Supplementary Note) explaining better the model and a new Supplementary Fig. 5 with model fitting.

I would like to be convinced that the elasticity of the cadherin molecules k plays a critical role in the story because it is not the first thing I would have guessed. The reviewer is correct. The parameter k influences the shape of the interface, as k is related to stiffness of E-cadherin bonds and the spatial density of these bonds. This is because E-cadherin bonds explicitly changes the force balance at the interface in the normal direction, as shown in the newly provided supplementary information on the model. The actual strain in the E-cad bonds, however, is determined by Eq. 2, and we see it is a combination of cortical stress and pressures. In support of the varying elasticity behaviour measured by parameter k , an interesting recent paper² strengthen the point that E-cadherin bonds exist at an equilibrium of different trans and cis interactions. This equilibrium results in clusters of different sizes and densities that are governed by forces sensed at the receptors. We added both points to the manuscript (pages 6-7 and 12).

Minor comments:

- It would have been easier to number each line We provide now all changes introduced to the manuscript as track-changes, which makes easier to identify and read them in the context of the text.

- The parameters of the model are randomly labeled alpha or a, including in Figure S5. We apologise for the oversight. It seems that there is an issue with symbol font from Latex-derived text or those found in Word. For consistence, we replaced all parameters with Symbol font in Word and revised Supplementary Fig. 6E, using a larger Symbol font in Illustrator.

- Works from like: Liu, Z. et al. Mechanical tugging force regulates the size of cell–cell junctions. Proc.Natl Acad. Sci. USA 107, 9944–9949 (2010)., Maruthamuthu, V., Sabass, B., Schwarz, U. S. & Gardel, M. L. Cell–ECM traction force modulates endogenous tension at cell–cell contacts. Proc. Natl Acad. Sci. USA 108, 4708–4713 (2011)., Tzeng et al PNAS January 31, 2012 109 (5) 15061511; <https://doi.org/10.1073/pnas.1106377109> have already described the interplay between cell shape and junction geometries and could be cited. These references report new important knowledge, and all three references were present in the text. In the original manuscript, Maruthamuthu et al. 2011 was already cited as reference 52 (now ref. 56), Li et al 2010 cited as reference 67 (now ref. 71) and Tseng et al was cited as reference 38 (now ref. 39).

- The sentence ending by “the product of number density and stiffness of the cadherin bonds” is unclear Thank you for raising this point. We revised the text to explain better what the parameter k means in page 6 and 7.

- Figure 4H should zoom on the results rather than have a large span on the y axis. Thanks for the suggestion. We modified the Y axis accordingly with revised Fig. 4H as requested. However, as shown in the table below, the experimental values of the fitting parameters across different shapes are very similar and enlarging the Y axis (Fig. 4H) would not be able to separate the symbols substantially.

Linearity CellType	Data (Extract from plots)			Prediction (From model)		
	○	□	△	○	□	△
Normal Cells	1.1263	1.1140	1.0789	1.12623	1.10891	1.10426

- In my compiled version the letters identifying each sub panel are gigantic. They could certainly be reduced to ease the reading of the figure. The font size of letters in each panel were reduced as requested.

- When the authors say: Having shown the distinct F-actin organization in the cytoplasm of more tensile neighbours, we inferred that higher stiffness of triangular cell pairs (Fig. 2A-B) would drive increased levels of E-cadherin at junctions to enable cells to sustain the increased mechanical load at cell-cell contacts. “Do they consider the tangential stress or the component of the stress orthogonal to the junction? As mentioned in the manuscript, in the cell doublet system, we consider that the stress at cell-cell contacts is mostly orthogonal or normal to the junctions, due to the absence of tricellular junctions and additional neighbours. The above statement is based on the SICM measurements at junctions and the distinct F-actin organization in more tensile cell pairs. We also considered the current assumptions in the literature that higher stress correlates with increased levels of receptors to reinforce junctions. We added further explanations in page 9 to make it clearer. Orthogonal and tangential stresses are also mentioned in page 11.

- In the sentence: “similar pattern of junction linearity with and with contractility inhibition” a word is missing. We apologise for the oversight. The phrase is now corrected in page 10.

- I do not quite understand what this sentence means.” Here, we demonstrate that cells modulate their intrinsic rheological properties and drive junction maturation by reorganizing the configuration and molecular organization of cell-cell contacts.” We agree the sentence is unclear and it was revised in page 11.

- In the sentence: “A cell autonomous process to increase intracellular pressure and cortical stiffness (without stretching or external forces...” I would have changed and in or Thanks for pointing this out. The phrase was corrected by replacing “and” for “or” in page 11.

- Why do you introduce the notion of viscoelasticity at the end of the paper when you never mentioned anything about viscous dissipation and treated the prob as a purely elastic problem? We revised the text to make it consistent throughout and emphasise the viscoelasticity concept in the results section.

- To derive the equation to measure the young modulus using SICM, do you assume that the medium is elastic? What Young modulus do you measure? That of the cortex? It seems likely that it is more the bending modulus that is measured that depends on the cortical tension. Could you elaborate her a bit to what part of the cell this modulus is associated to? As the reviewer inferred, the equation to measure Young’s modulus assumes an elastic medium. It was defined by Rheinlaender & Schäffer³ as a measure of cortical stiffness. We apologize that the information and reference were not clearly provided in the main text (only in the supplementary material). We extended the explanation of the equation in a revised Supplementary Methods (page 6) and modified the main text to explain with more detail what was measured (page 5).

Reviewer #2

The reviewer finds our results “interesting, and uncover non-trivial relations between the geometry and the junction's shape and cell elasticity, and the local organization of the actin, myosin and cadherin adhesion bonds.” He/she raises the following points to be addressed:

1) This is relevant citation with respect to the fluctuations in the monolayer: Fodor, É., Mehandia, V., Comelles, J., Thiagarajan, R., Gov, N. S., Visco, P., ... & Riveline, D. (2018). Spatial fluctuations at vertices of epithelial layers: quantification of regulation by rho pathway. *Biophysical journal*, 114(4), 939-946. We are aware of the above citation that reports fluctuation of vertices in an epithelial

monolayer. However, our model does not contain junction vertices (i.e., tricellular junctions) and for this reason the reference was not cited. Yet, the paper is relevant to highlight that the fluctuations rely on contractile status of the monolayer. The paper is now added to the discussion (page 11) as reference 53.

2) By speaking about "viscoelasticity" of the junction they give the impression its a passive material property, while it relies on ATP-consuming processes related to acto-myosin contractility and actin polymerization. This should be explained. This is a misunderstanding. We apologise if the text was unclear. The adaptive mechano-responses upon change in viscoelasticity relies on ATP-derived energy, either by remodelling of the cytoskeleton or intracellular transport to modulate levels of receptors at junctions. We revised the text to clarify the energy-dependent process in pages 4 and 12.

3) I have a general problem: isnt a 2-cell system inherently less stable, unlike a 3-cell system, which has a very stable tri-cellular junction? wouldn't it be more revealing and relevant for tissues and monolayers to study a 3-cell system? Our aim was to design a minimal system to dissect the essential requirements for junction maturation and stability. Such minimal system is much more pliable and easier to dissect key signalling events underlying the mechanical responses. Although a two-cell system could be seen as more unstable, the Junction Unit model displays various classical parameters of junction configuration, compaction and myosin-dependent F-actin remodelling as seen in epithelial monolayers. While the maturation process in epithelial tissues is indeed more complex, the cell doublet model contributes to our understanding of the adaptive responses of intrinsic cell rheology.

We can now add further complexity by confining three cells and analysing how asymmetry, geometric shape and mechanical responses influence tricellular junctions. Further complexity will add to our framework to dissect the tensional contributions of additional neighbours, morphological polarization with the concomitant remodelling, function and segregation of additional adhesive systems. However, these analyses are beyond the scope of the current manuscript.

4) page 5, English typos: "ability of attach to each other", "each neighbouring interacts", page 6: "cadherin bondsWe assume". More typos at the top of page 8. Apologies for the oversight. We revised the text throughout, and different typos were corrected.

5) Does the local Young's modulus correlate with the local fluorescene intensity of actin, or myosin, or actin+myosin? can they show such a comparison. The SICM measurements were done in live cells and the F-actin quantification was performed on fixed samples (please see also our reply to reviewer 3 point 2). Thus, a direct comparison in the same cells has not been possible. There is a well-established correlation of stiffness and actin organization in the literature, as cytoskeleton perturbing drugs such as cytochalasin softens the cells. Our stiffness profiles together with the cytoskeletal organization of confined cell doublets demonstrate that similar range of Young's moduli values can be obtained at remarkably distinct F-actin structures, with different filament organization and molecular composition, such as lamellae, vertices or junctions. We argue that more in-depth analyses should be performed to account for those previously unappreciated actin structures that contribute to localised stiffness. In our manuscript, we begun to address these with our modelling and by cross-correlation between Young's Moduli, phosphorylated myosin light chain levels (PMLC) and density of both receptors and F-actin at junctions.

6) Where in equations (1) do we see that if the junction is longer it can allow for more adhesion sites and therefore provides a larger negative adhesion energy that can be a driver to increase the length? I dont see this adhesion-derived term in these equations. The Reviewer is correct. Eq. 1 is force balance only; it does not contain free energy driving force from forming more interfacial E-cadherin bonds. Thus, the model assumes a particular distribution and number of E-cadherin bonds, as reflected in the k parameter. In principle, we can also vary the E-cadherin density and number. However, we take k from the experimental measurement of E-cad intensity. The question of interfacial contour length is also potentially complex and may also depend on the cortical actin dynamics. This

issue is related to how k is determined and is beyond the scope of this paper. Please see also our reply to point 8 below.

7) On page 9 they write: "Yet, inhibition of intracellular stiffness did not alter the strong correlation between area and volume of each cell". Why should it? We realise that the point was not well explained. There are not many publications comparing stiffness, cell volume and area. The correlations obtained herein are distinct from the few reports available. Our data on confined cell doublets show a positive correlation between volume and cell area of each cell. However, an interesting paper from the Weitz lab shows that a confined single cell has an inverse correlation between cell volume and cell area or cortical stress.⁴ Such correlations are maintained upon relaxation of unconstrained single cells, where a higher cell area (spreading) is accompanied by reduced cell volume.⁴ In contrast, during *Drosophila* dorsal closure and monolayer invagination, increased contraction correlates with reduced volume,⁵ or cell volume is conserved as cell elongates, respectively.⁶ We surmise that correlations among biophysical parameters depend on the biological model, confinement and experimental conditions. The text was re-written and relevant citation⁴ (new ref. 47 in main text) was added in page 10.

Reviewer Fig.1: A, Diagram showing the measurement of junction length (cadherin staining) h , which not always extend to the borders of the micropatterns. B, Inverted images in grey scale and the undulated junction length highlighted in blue. C, quantification of the junction length of cell doublets in different geometries. Average length is progressively reduced in squared and triangular doublets (as shown in manuscript Fig.3 G).

8) They write: "In contrast, the density of E-cadherin at cell-cell contacts increased with contraction inhibition of cell pairs on squares and triangles, but not on circular doublets (Fig. 8B). Upon relaxation, junctions became significantly more curvilinear in triangular-shaped cell pairs (higher linearity index, Fig. 8C)." Isn't this a possible indication that: The increase in cadherin at the junction on the triangular shapes may drive the increase in junction length, to allow for larger "wetting" of the two cells. The reviewer raises an important point. As described in the Results section page 6, the measured length of junctional E-cadherin staining (i.e., fluorescence signal at junctions) is on average significantly shorter in triangular cell doublets (Fig.3G and as shown in Reviewer Fig.1). Yet, the ability of cadherin receptors to localise at the interface between neighbouring cells is similar in different geometries. About 80% of the contacting interface length is occupied by the receptors at steady state in all samples (coverage index; Reviewer Fig. 2A-B and manuscript Fig.3H). These results suggest that the length of E-cadherin staining is not always the same length as the interfacial membrane.

When cell contractility is perturbed by Y27632 treatment (Reviewer Fig. 2C), the more undulated junctions of triangular doublets have a significant reduction in coverage index. In spite of the increased undulation of the contacting interface upon relaxation, there is no concomitant increase in recruitment or spread of receptors at adhesion sites.

Thus, cadherin receptors cover less of the available contacting interface length (Reviewer fig. 2C) but have brighter clusters and intensity (Fig. 8B). Clearly, future studies on cell

Reviewer Fig.2: A, Coverage index is defined here as the percentage of the available contacting interface that is covered by staining of cadherins. B, Coverage index of the single junction of cell pairs on different geometries (as shown in manuscript Fig.3H). C, Relaxation of cell pairs promotes a significant reduction of the coverage index in triangular-shaped doublets.

relaxation, membrane turnover and cadherin receptor dynamic delivery/retrieval to and from junctions will confirm our prediction. We raised the above points in the manuscript (page 10) and included Reviewer Fig. 2C as Supplementary Fig. 6E.

9) Are the cells on the circular islands rotating? this could also affect the shape of the junction. The reviewer is correct: cell pairs in different geometries rotate with distinct properties. We developed suitable software to measure the movement of confined cell doublets in video-time lapse and the data will be reported in a separate manuscript (*in preparation*). The current manuscript offers a snapshot of a dynamic interplay between cell doublets. The quantitative analyses of various parameters reflect snapshots of the most frequent and preferred behaviour of cell pairs on each geometric shape. The above rationale was added to page 11.

10) What is so special about shapes with sharp corners? they seem to be sites from which cell-spanning stress-fibers emanate. Are these indeed the main drivers of the differences? How does the cell-cell junction change when the SF at the sharp corner are laser-ablated, for example? This is a very interesting question that we have been unable to address with our current model. We predict that laser ablation would cause cell retraction from the micropattern vertices (as seen with junctions) and thus may not be helpful to address this question. We envisage that the design of additional micropattern shapes will help to dissect the contribution of vertices with sharp versus shallower angles.

11) In Fig.1B I think that in the bottom image the text is exchanged. Apologies for the oversight. Text is now corrected in page 13.

12) Fig.2A: In this panel the circular cells seem stiffer than those on the square, right? this is against the reported trend. Thank you for raising this point. There is indeed variability in the elasticity map of the different cell pairs. We have changed the image of the elasticity and topography profiles of square doublets in Fig.2A to be more consistent with the range of the measurements seen in Fig. 2B.

13) Fig.3: Give the distribution of the ratio between the two cells for each geometry, to clearly see how the geometry affects the asymmetry between the neighbours. We thank the reviewer for the suggestion and the average data (rather than matched neighbours) is shown in Reviewer Figure 3. We have incorporated the result in Supplementary Fig. 4G-H and added explanations to Results page 6. The larger discrepancy in cell area between two neighbours is observed in the triangular-shaped cell pairs compared to circular or squared cell doublets (Reviewer Fig.3A). The average cell area ratio is similar for cell pairs on different geometric shapes (Reviewer Fig.3B).

14) Fig.3E: Seems that the junction position always chose approximately the longest straight path: equator in the circle, diagonal in the square and the altitude in the triangle. Is this to maximize the contact area and therefore maximize the adhesion energy? We realise that Fig.3E can be misleading in the way it was presented. We apologize for the mistake and oversight. We revised the data to show the raw data quantified (to enable comparison across all shapes) and added the probability of a random distribution of a junction at 50 intervals at the sides or spanning a vertex of squared and triangular cell pairs. The revision does not change the conclusion that there is higher probability for junctions to be found at the vertex areas (i.e., within 20° degrees for squares or 15° degrees for triangles). We have altered the graph in Figure 3E, added more explanations in the Supplementary Methods (page 7-8) and in the Results section (page 6).

Reviewer Fig.3: The cell area of each neighbouring cell on a micropattern was calculated and expressed as the difference among the areas of the larger and smaller cell (A) or as the ratio between smaller and larger area values (B)

We argue that the preference of junction positions in squared and triangular shapes will be better addressed in the context of dynamic movement that cell doublets have on the distinct geometries (rather than the snapshots available). We are currently developing the quantitative imaging tools for these analyses and the data will be reported as a separate manuscript.

Reviewer #3

The reviewer values our experimental approach, quality of the data and how they support our conclusions and provide a more complete understanding of junction maturation. The main concern of the reviewer is about how we performed the SICM, which was mistakenly understood as performed in fixed cells. Our detailed reply is below.

1. SICM is a unique tool that offers high-resolution topography and stiffness measurements of the soft eukaryotic cell membrane without mechanical contact. I'm glad the authors used SICM instead of AFM, which is a more established technique but may not be able to detect lower stiffness regions. What's is the sensitivity of the stiffness measurement with the SICM method used in this study? The authors should consider performing some AFM experiments to compare and validate the SICM stiffness measurements? The stiffness measurements in our work uses the method first described by Rheinlaender and Schäffer.³ The method has been validated by the same group to confirm it provides values in agreement with the more established AFM.³ In their original work, Rheinlaender and Schäffer describe that the sensitivity of the SICM method is effectively determined by the pressure applied to the pipette (p0) and can measure elastic modulus as low as 10 Pa. Furthermore, various other studies have compared the AFM and SICM with respect to morphology imaging, height, and stiffness (e.g.,⁷⁻¹¹). This information has been added to the manuscript Supplementary Methods (page 6).

2. A major concern is the use of fixed cells in this study. Paraformaldehyde (PFA) crosslinks the many molecules and structures in the cell. Therefore, I would expect an increased stiffness of the measurements compared with live cells. It could be that the use of fixed cells still shows similar relative behaviors, but the absolute numbers are almost certainly wrong. For this manuscript to be acceptable, the authors would need to either repeat the measurements with live cells, or at the least show comparison measurements between live and fixed cells. Depending on these results, perhaps the authors can explain why using fixed cells is acceptable. This is a misunderstanding. All SICM experiments in our manuscript were done with live cells. The reviewer is correct about potential alterations by fixation. Previous work using AFM show that glutaraldehyde fixation stiffens endothelial cells,⁷ and that fixation improves the topography imaging of soft subcellular structures by AFM (i.e. microvillae).¹⁰ In contrast, SICM topography resolution is more refined and stable during acquisition and is not substantially improved by fixation.^{8, 9, 11} As fixed cells were not used in our SICM experiments, and considering the various comparisons published on fixed and live cells,⁷⁻¹¹ we posit that amendment of the text with relevant citations will be sufficient to clarify the issue to readers.

3. Why did the authors use PFA 3% fixation? Was it the one they expect to affect the measurement the least? If so, why? If the authors continue the path with the fixed cells, I propose they test several fixation conditions. PFA 4%, 2%, w/o triton. Triton may also change the mechanical properties measured as small holes perfuse and degrade the membrane. The authors should show the effect of the fixation on the measurements. As stated above, all our measurements were performed with live cells; the fixation with 3% paraformaldehyde was only used for immunofluorescence. We apologise for the confusion and the methods have been revised to clarify that only live cells were used for SICM experiments (Supplementary Methods pages 5-6) and further explanations added to the results (page 5). We respectfully argue that, as fixation was not used in the SICM experiments in our manuscript, the concern of potential skewing the stiffness measurements is not applicable to the data presented.

References:

1. Takeichi, M. Dynamic contacts: rearranging adherens junctions to drive epithelial remodelling. *Nature Rev. Mol. Cell Biol.* **15**, 397–410 (2014).
2. Chen, Y., Brasch, J., Harrison, O.J. & Bidone, T.C. Computational model of E-cadherin clustering under force. *Biophys. J.* **120**, 4944-4954 (2021).
3. Rheinlaender, J. & Schäffer, T.E. Mapping the mechanical stiffness of live cells with the scanning ion conductance microscope. *Soft Matter* **9**, 3230 (2013).
4. Guo, M. *et al.* Cell volume change through water efflux impacts cell stiffness and stem cell fate. *Proc. Natl. Acad. Sci. USA* **114**, E8618-E8627 (2017).
5. Saias, L. *et al.* Decrease in Cell Volume Generates Contractile Forces Driving Dorsal Closure. *Dev. Cell* **33**, 611-621 (2015).
6. Gelbart, M.A. *et al.* Volume conservation principle involved in cell lengthening and nucleus movement during tissue morphogenesis. *Proc. Natl. Acad. Sci. USA* **109**, 19298-19303 (2012).
7. Braet, F., Rotsch, C., Wisse, E. & Radmacher, M. Comparison of fixed and living liver endothelial cells by atomic force microscopy. *Applied Physics A: Materials Science & Processing* **66**, S575-S578 (1998).
8. Rheinlaender, J., Geisse, N.A., Proksch, R. & Schaffer, T.E. Comparison of scanning ion conductance microscopy with atomic force microscopy for cell imaging. *Langmuir* **27**, 697-704 (2011).
9. Ushiki, T., Nakajima, M., Choi, M., Cho, S.J. & Iwata, F. Scanning ion conductance microscopy for imaging biological samples in liquid: a comparative study with atomic force microscopy and scanning electron microscopy. *Micron* **43**, 1390-1398 (2012).
10. Seifert, J., Rheinlaender, J., Novak, P., Korchev, Y.E. & Schaffer, T.E. Comparison of Atomic Force Microscopy and Scanning Ion Conductance Microscopy for Live Cell Imaging. *Langmuir* **31**, 6807-6813 (2015).
11. Kim, S.O., Kim, J., Okajima, T. & Cho, N.J. Mechanical properties of paraformaldehyde-treated individual cells investigated by atomic force microscopy and scanning ion conductance microscopy. *Nano Conv.* **4**, 5 (2017).

REVIEWER COMMENTS

Reviewer #1 (Remarks to the Author):

I congratulate the authors for the good work they made to clarify their experimental protocols and theoretical framework. More Generally I found the paper quite interesting.

Reviewer #2 (Remarks to the Author):

Reading the authors' responses to my questions, I am very confused. I don't think this work can be published in Nat Comm.

1) The model as presented in the paper is very simple, and deals with static force balance at the junction. As such, it can only describe an interface that bulges into one of the cells, due to a pressure imbalance. They write that "discrete, localized variations of pressure from cell 1 or cell 2 leads to an undulated or "wavy" contacting interface.", but what can lead to these localized pressure changes ?

Since, as they answer to my previous report, the cells in the circular pattern are moving, and this most likely leads to the wavy interface pattern. The difference between the cell-cell interface shape on the circular and triangular patterns may therefore crucially depend on the dynamics, which they do not treat at all.

See experiments and theory, for example:

Huang S, Brangwynne CP, Parker KK, Ingber DE (2005) Symmetry-breaking in mammalian cell cohort migration during tissue pattern formation: Role of random-walk persistence. *Cell Motil Cytoskeleton* 61(4):201–213.

Brangwynne C, Huang S, Parker KK, Ingber DE, Ostuni E (2000) Symmetry breaking in cultured mammalian cells. *In Vitro Cell Dev Biol Anim* 36(9):563–565.

These are very relevant references that they do not cite, and have very important implications for how they model and analyze their data:

Albert, Philipp J., and Ulrich S. Schwarz. "Modeling cell shape and dynamics on micropatterns." *Cell adhesion & migration* 10.5 (2016): 516-528.

Albert, Philipp J., and Ulrich S. Schwarz. "Dynamics of cell ensembles on adhesive micropatterns: bridging the gap between single cell spreading and collective cell migration." *PLoS computational biology* 12.4 (2016): e1004863.

Albert, Philipp J., and Ulrich S. Schwarz. "Dynamics of cell shape and forces on micropatterned substrates predicted by a cellular Potts model." *Biophysical journal* 106.11 (2014): 2340-2352.

2) They do not show examples of the solutions for the inter-cell gap $l(s)$, which determine the cadherin contribution to the force balance. This will help to assess this contribution.

Reviewer #3 (Remarks to the Author):

The reviewers have properly addressed my concerns in the revised manuscript

Response to Reviewers rebuttal

We thank Reviewers 1 and 3 for the compliments to the revised manuscript and for finding our manuscript now acceptable for publication. We answer below the questions raised by Reviewer 2 with regards to the model used in the manuscript. Modifications in the manuscript are found in red font.

Reviewer 2

1) The model as presented in the paper is very simple, and deals with static force balance at the junction. As such, it can only describe an interface that bulges into one of the cells, due to a pressure imbalance. They write that "discrete, localized variations of pressure from cell 1 or cell 2 leads to an undulated or "wavy" contacting interface.", but what can lead to these localized pressure changes?

Using static images, the model has been informative in providing new insights into the mechanisms of junction configuration and maturation. Global changes in hydrostatic pressure among neighbouring cells would bulge the whole contacting interface. We predict that external pressure from a neighbour triggers a localized deformation of the contacting interface by two potential mechanisms: (i) the positioning of the bulky nuclei whose proximity to the junction that impinge its configuration (ii) an adaptive biomechanical response that coordinates spatially restricted cortex fluidization and fluctuations of intracellular hydrostatic pressure. Such precise spatial and temporal control is consistent with previous publications reporting oscillatory behaviour of signalling at the membrane or cytoplasm and a heterogeneity of markers along contacting surfaces:

- Cell blebbing during ameboid motility or during apoptotic cell death, where discrete areas of the cell surface protrude while other membrane areas contract and retract (e.g., ¹⁻³);
- Pulse or flashes of activation of signalling across the surface of epithelial monolayers that accompany contractile events during morphogenesis;⁴⁻⁹
- The selective remodelling of individual junctions in an epithelial monolayer that can reduce or elongate thereby changing the number of neighbours and generating distinct folding and patterns during organogenesis;¹⁰⁻¹⁴
- The known heterogeneity of adhesive receptors and other signalling molecules along the vertical axis of epithelial junctions (top to bottom)¹⁵⁻¹⁷ and the horizontal axis (tricellular versus bilateral junctions),¹⁸⁻²⁰ which provide distinct spatial clues and distinct biomechanical properties.

It is feasible that similar spatial and temporal control exists for the mechanical sensing and ion channels at the border between two cells. However, the precise molecular mechanisms for the undulated or wavy interface are beyond the scope of the current manuscript. We added further explanations in the text (page 12).

Since, as they answer to my previous report, the cells in the circular pattern are moving, and this most likely leads to the wavy interface pattern. The difference between the cell-cell interface shape on the circular and triangular patterns may therefore crucially depend on the dynamics, which they do not treat at all. See experiments and theory, for example:

Huang S, Brangwynne CP, Parker KK, Ingber DE (2005) Symmetry-breaking in mammalian cell cohort migration during tissue pattern formation: Role of random-walk persistence. *Cell Motil Cytoskeleton* 61(4):201–213.

Brangwynne C, Huang S, Parker KK, Ingber DE, Ostuni E (2000) Symmetry breaking in cultured mammalian cells. *In Vitro Cell Dev Biol Anim* 36(9):563–565.

Apologies for not been clearer in our previous response. In our model, all cell pairs are moving in all geometries: circular, squared and triangular. We have three points to make. First, the undulated phenotype occurs independently of cell motility and results from alterations of contractile and tensional properties of cell cortex. The "wavy" cell-cell contact phenotype has been identified using static images of epithelial monolayers (Reviewer Fig.1) and has been associated with relaxed contractility at junctions regulated by Cdc42 signalling²¹ or ROCK activity.²² We build from this pioneering work to model additional biomechanical properties that contribute to such phenotype. Second, we take on board the reviewer's point on the dynamic behaviour of cells. However, modelling of static images is frequently done in publications. All biological processes are dynamic, but this does not invalidate our current findings and modelling, as the time scales are very different. Keratinocyte cell pairs on

confined spaces as described in our manuscript move very slowly at 0.1 microns/min, and after 90 min, total displacement of their centroids is about 13 microns (for comparison, a keratinocyte cell is 20-40 μm long). In contrast, interface shape movement and remodelling are faster, and the mechanical equilibrium would be established in seconds. Thus, here we focus on the static shape of the cell-cell junction and the faster events that underpin their remodelling and maturation.

[redacted]

Finally, the Junction Unit system is similar to what published previously by Ingber's group.²³ However, the authors model cell motility, which is distinct to our questions on the impact of intracellular rheology properties on junctions. The cell type used is distinct (capillary endothelia) and displays a very floppy cell-cell contact (Ying-Yang shape).²³ These cells move in a rotational motion, roughly perpendicular the interface. The migration force should have relatively little impact on the interface shape: a few microns above the substratum, but this is approximate (in keratinocytes, junctions are positioned about 6-7 microns above ECM). In any case, we only look at the static case and established an important framework to investigate the relationships between cell rheology and neighbour interactions. The contribution of the dynamic cell status is explored in depth in a separate study. We added further explanations in the revised text page 6.

These are very relevant references that they do not cite, and have very important implications for how they model and analyze their data:

Albert, Philipp J., and Ulrich S. Schwarz. "Modeling cell shape and dynamics on micropatterns." *Cell adhesion & migration* 10.5 (2016): 516-528.

Albert, Philipp J., and Ulrich S. Schwarz. "Dynamics of cell ensembles on adhesive micropatterns: bridging the gap between single cell spreading and collective cell migration." *PLoS computational biology* 12.4 (2016): e1004863.

Albert, Philipp J., and Ulrich S. Schwarz. "Dynamics of cell shape and forces on micropatterned substrates predicted by a cellular Potts model." *Biophysical journal* 106.11 (2014): 2340-2352.

We thank the reviewer for pointing out these references and the insightful information that they contain. The papers from Schwartz lab use phenomenological cellular Potts model and Vertex model.²⁴ In the papers, the authors model cell spreading over specific thin regions coated with substratum (L, V, Y or X shape),²⁴⁻²⁶ rather than changes in cell area and interface contact shaped by both neighbours and geometric confinement. While not directly comparable to our model system, we cited two of the references in the revised text (page 6) to highlight their contribution to the field and state the differences of the biological questions addressed in their paper and in our study.

2) They do not show examples of the solutions for the inter-cell gap $l(s)$, which determine the cadherin contribution to the force balance. This will help to assess this contribution.

We thank the reviewer for the suggestion. The modelling for the intergap $l(s)$ contribution to the force balance is shown below (Reviewer Fig. 2). The new data (right panel) is incorporated into revised Supplementary Fig. 5E and relevant text is introduced in pages 8 and 9.

Reviewer Fig.2: Modelling the contribution of intergap parameter $l(s)$ to the force balance at the contacting interface. Panel on the right shows controls with the parameter K (cadherin strength) set at 2kPa. Panel on the right are the results with K set at 1.5 kPa.

References:

1. Colemann, M.L. *et al.* Membrane blebbing during apoptosis results from caspase-mediated activation of ROCK 1. *Nature Cell Biol.* **3**, 339-345 (2001).
2. Li, A. & Machesky, L.M. Rac1 drives melanoblast organization during mouse development by orchestrating pseudopod-driven motility and cell-cycle progression. *Small GTPases* **3**, 115-119 (2012).
3. Naji, L., Pacholsky, D. & Aspenstrom, P. ARHGAP30 is a Wrch-1-interacting protein involved in actin dynamics and cell adhesion. *Biochem. Biophys. Res. Comm.* **409**, 96-102 (2011).
4. Stephenson, R.E. *et al.* Rho Flares Repair Local Tight Junction Leaks. *Dev. Cell* **48**, 445-459 e445 (2019).
5. Cavanaugh, K.E., Staddon, M.F., Munro, E., Banerjee, S. & Gardel, M.L. RhoA Mediates Epithelial Cell Shape Changes via Mechanosensitive Endocytosis. *Dev. Cell* **52**, 152-166 e155 (2020).
6. Arnold, T.R., Stephenson, R.E. & Miller, A.L. Rho GTPases and actomyosin: Partners in regulating epithelial cell-cell junction structure and function. *Exp. Cell. Res.* **358**, 20-30 (2017).
7. Coravos, J.S., Mason, F.M. & Martin, A.C. Actomyosin Pulsing in Tissue Integrity Maintenance during Morphogenesis. *Trends Cell Biol.* **27**, 276-283 (2017).
8. Bement, W.M. *et al.* Activator-inhibitor coupling between Rho signalling and actin assembly makes the cell cortex an excitable medium. *Nature Cell Biol.* **17**, 1471-1483 (2015).
9. Reyes, C.C. *et al.* Anillin regulates cell-cell junction integrity by organizing junctional accumulation of Rho-GTP and actomyosin. *Cur. Biol.* **24**, 1263-1270 (2014).
10. Bertet, C., Sulak, L. & Lecuit, T. Myosin-dependent junction remodelling controls planar cell intercalation and axis elongation. *Nature* **429**, 667-671 (2004).
11. Chandran, R., Kale, G., Philippe, J.M., Lecuit, T. & Mayor, S. Distinct actin-dependent nanoscale assemblies underlie the dynamic and hierarchical organization of E-cadherin. *Cur. Biol.* **31**, 1726-1736 e1724 (2021).
12. Lecuit, T. & Yap, A.S. E-cadherin junctions as active mechanical integrators in tissue dynamics. *Nature Cell Biol.* **17**, 533-539 (2015).
13. Martin, A.C., Gelbart, M., Fernandez-Gonzalez, R., Kaschube, M. & Wieschaus, E.F. Integration of contractile forces during tissue invagination. *J. Cell Biol.* **188**, 735-749 (2010).
14. Martin, A.C., Kaschube, M. & Wieschaus, E.F. Pulsed contractions of an actin-myosin network drive apical constriction. *Nature* **457**, 495-499 (2009).
15. Mack, N.A. *et al.* beta2-syntrophin and Par-3 promote an apicobasal Rac activity gradient at cell-cell junctions by differentially regulating Tiam1 activity. *Nature Cell Biol.* **14**, 1169-1180 (2012).
16. Vassilev, V., Platek, A., Hiver, S., Enomoto, H. & Takeichi, M. Catenins Steer Cell Migration via Stabilization of Front-Rear Polarity. *Dev. Cell* **43**, 463-479 e465 (2017).
17. Peglion, F., Llense, F. & Etienne-Manneville, S. Adherens junction treadmill during collective migration. *Nature Cell Biol.* **16**, 639-651 (2014).
18. Bosveld, F. *et al.* Epithelial tricellular junctions act as interphase cell shape sensors to orient mitosis. *Nature* **530**, 495-498 (2016).
19. Oda, Y., Otani, T., Ikenouchi, J. & Furuse, M. Tricellulin regulates junctional tension of epithelial cells at tricellular contacts through Cdc42. *J. Cell Sci.* **127**, 4201-4212 (2014).
20. Salomon, J. *et al.* Contractile forces at tricellular contacts modulate epithelial organization and monolayer integrity. *Nature Comm.* **8**, 13998 (2017).
21. Otani, T., Ichii, T., Aono, S. & Takeichi, M. Cdc42 GEF Tuba regulates the junctional configuration of simple epithelial cells. *J. Cell Biol.* **175**, 135-146 (2006).
22. Priya, R. *et al.* ROCK1 but not ROCK2 contributes to RhoA signaling and NMIIA-mediated contractility at the epithelial zonula adherens. *Mol. Biol. Cell* **28**, 12-20 (2017).
23. Huang, S., Brangwynne, C.P., Parker, K.K. & Ingber, D.E. Symmetry-breaking in mammalian cell cohort migration during tissue pattern formation: role of random-walk persistence. *Cell Motil. Cytosk.* **61**, 201-213 (2005).
24. Albert, P.J. & Schwarz, U.S. Dynamics of cell shape and forces on micropatterned substrates predicted by a cellular Potts model. *Biophys. J.* **106**, 2340-2352 (2014).
25. Albert, P.J. & Schwarz, U.S. Modeling cell shape and dynamics on micropatterns. *Cell Adh. Migr.* **10**, 516-528 (2016).
26. Albert, P.J. & Schwarz, U.S. Dynamics of Cell Ensembles on Adhesive Micropatterns: Bridging the Gap between Single Cell Spreading and Collective Cell Migration. *PLoS Comp. Biol.* **12**, e1004863 (2016).

REVIEWERS' COMMENTS

Reviewer #1 (Remarks to the Author):

To my opinion the paper is ready for publication and the last answers to referee 2 are satisfactory.